# Bayesian inference of neuronal assemblies

Giovanni Diana●*, Thomas T. J. Sainsbury●, Martin P. Meyer●*

Center for Developmental Neurobiology & MRC Center for Neurodevelopmental Disorders, King's College London, Guy's Hospital Campus, London, United Kingdom

* giovanni.diana@kcl.ac.uk (GD); martin.meyer@kcl.ac.uk (MPM)

## Abstract

In many areas of the brain, both spontaneous and stimulus-evoked activity can manifest as synchronous activation of neuronal assemblies. The characterization of assembly structure and dynamics provides important insights into how brain computations are distributed across neural networks. The proliferation of experimental techniques for recording the activity of neuronal assemblies calls for a comprehensive statistical method to describe, analyze and characterize these high dimensional datasets. The performance of existing methods for defining assemblies is sensitive to noise and stochasticity in neuronal firing patterns and assembly heterogeneity. To address these problems, we introduce a generative hierarchical model of synchronous activity to describe the organization of neurons into assemblies. Unlike existing methods, our analysis provides a simultaneous estimation of assembly composition, dynamics and within-assembly statistical features, such as the levels of activity, noise and assembly synchrony. We have used our method to characterize population activity throughout the tectum of larval zebrafish, allowing us to make statistical inference on the spatiotemporal organization of tectal assemblies, their composition and the logic of their interactions. We have also applied our method to functional imaging and neuropixels recordings from the mouse, allowing us to relate the activity of identified assemblies to specific behaviours such as running or changes in pupil diameter.

**Data Availability Statement:** Our C++ software implementing the inference method described in this work together with custom R-scripts to reproduce the analysis are publicly available on the GitHub repository https://github.com/giovannidiana/BINE. The recordings from the

## Author summary

Characterization of the structure and dynamics of population activity can provide insight into how computations are distributed within neural networks. Here we develop a new statistical method to describe patterns of synchronous activity in neural population recordings. Our method can accurately describe how neurons are organized into co-active populations (assemblies) and reveals dynamic features of assemblies such as their firing pattern, firing duration, and the degree of synchronous and asynchronous firing of their constituent neurons. We demonstrate how our technique can be used to dissect complex neuronal population recording data into its behaviorally relevant components.

This is a *PLOS Computational Biology* Methods paper.

zebrafish tectum are publicly available on figshare: https://doi.org/10.6084/m9.figshare.9994799.v1.

**Funding:** Martin Meyer was financially supported by a Wellcome Trust Investigator Award MPM:204788/Z/16/Z (https://wellcome.ac.uk). The funders had no role in study design, data collection and analysis, decision to publish, or preparation of the manuscript.

# Introduction

Donald Hebb proposed that neural circuits are specifically built to enable the generation of sequential activity patterns and that through synaptic plasticity, coactive neurons would be bound together into Hebbian assemblies—groups of neurons that tend to be coactive. Recent advances in multineuronal recording techniques such as calcium imaging and neuropixels probes have revealed that a hallmark of population activity is indeed the organization of neurons into assemblies [1–6]. Furthermore, activation of specific assemblies has been shown to correlate with diverse aspects of brain function from encoding of sensory stimuli to generation of motor output [6–8]. Thus, neural assemblies are increasingly believed to be the units of brain computation [9, 10]. The characterization of assembly structure and dynamics may therefore provide crucial insights into how brain computations are distributed across neural networks.

However, quantitatively identifying and extracting features of assemblies based on population firing statistics is challenging. Neurons can fire independently of the assembly of which they are a member and not all neurons within an assembly are recruited when the assembly fires. Assemblies can also exhibit temporal overlap or hierarchical organization, which makes it difficult to distinguish them from one another. Neurons may not participate in assembly activity at all, in which case their activity contributes to noise in the data. Finally, assemblies themselves may be of different sizes, have different activity rates, and neurons within them may be recruited over different timescales.

In the absence of a quantitative definition of neuronal assemblies, previous works applied heuristic methodologies such as dimensionality reduction techniques (e.g. principal component analysis, PCA) combined with factor analysis and graph-theoretic methods to characterize assemblies [5, 6, 8, 11–15]. Common to all these approaches is that they generate an abstract representation of the data. For example PCA describes the data in terms of principal components while spectral clustering requires embedding activity patterns in a network (see Materials and methods for details). These approaches present significant problems when trying to determine the correct number of assemblies present in neuronal recordings. PCA-based methods assume that this number is equal to the number of principal components required to describe the variance of the data that cannot be explained by chance according to theoretical bounds or null models obtained by shuffling the data. Graph-theoretic approaches rely on the fidelity of a graph representation and the equivalence of neuronal assemblies to network communities in the graph. However the construction of the representative network requires several arbitrary decisions (number of nearest neighbours and similarity matrix, for instance) which are not directly related to biological features. In instances where the data are clearly structured into synchronous events and there is minimal noise these methods can perform well [16]. However, these methods can lead to erroneous conclusions about how neurons are organised into assemblies as noise and data complexity increase [16]. The poor performance of some of the current analysis pipelines is perhaps not surprising given that standard clustering algorithms are mainly used as exploratory tools and not for statistical inference. Importantly, because information about the level of structure in the data is generally not known a priori, the accuracy of these estimations is also not known. Furthermore, because these methods are not based on statistical inference, their conclusions cannot be supported by statistical evidence.

In order to address these problems we have developed a method that is specifically tailored to neuronal data where data features such as noise, level of within-assembly synchrony and assembly activity are directly estimated from the data and simultaneously used to cluster neurons into assemblies. We have developed a hierarchical model of neuronal activity and used Bayesian inference to fit the observed synchrony among neurons with our model. The method

allows us to obtain information about assembly structure and dynamics from the recorded population activity, without the need of dimensionality reduction or similarity measures. By employing statistical inference, neurons are grouped into assemblies based on the most likely scenario that is consistent with the data and the model assumptions. Our method generates probabilistic estimates of assembly ON/OFF states over time and within-assembly statistical features such as synchronous and asynchronous firing probabilities. A common issues in all existing methods is the difficulty of determining how many assemblies are present in neuronal population activity data. In our Bayesian method, this number is also estimated from the data by introducing the Dirichlet process, which allows us to treat the number of assemblies as any other model parameter.

In our method all the assumptions underlying the definition of neuronal assemblies are transparent and explicitly stated as parameters in the model. Furthermore, because our model is grounded on Bayesian inference, all our conclusions are supported by statistical evidence. Earlier works have used Bayesian estimation of neural connectivity in the context of realistic models of neural dynamics such as the leaky integrate-and-fire model [17, 18]. However, the complexity of these models introduces scalability issues when applied to large datasets.

To determine the range of applicability of our method we tested it on simulated data and compared its performance against existing techniques over a broad range of sample size, assembly number and assembly features. We demonstrate that (1) our method outperforms existing techniques over a broad range of conditions and (2) our method provides optimal inference when testing data are simulated from the same generative model used for inference.

Using our method we have characterized neuronal assemblies from large-scale functional imaging data obtained from zebrafish tectum and mouse visual cortex, and from a neuropixels probe located in mouse visual cortex, hippocampus and thalamus [19]. We demonstrate that our method can provide a statistically rigorous approach to identifying assemblies, their intrinsic dynamics, interactions and coordinated activation and deactivation during behavior.

## Results

### Bayesian statistics

Given a model which describes observations in terms of a set of unobserved (latent) features $Z$ and model parameters $\theta$, the aim of Bayesian statistics is to quantify the probability distribution of parameters and latent variables conditional to the data using the Bayes' theorem

$$P(Z, \theta | \text{data}) \propto \overbrace{P(\text{data}, Z | \theta)}^{\text{data likelihood}} \times \overbrace{P(\theta)}^{\text{prior}} \tag{1}$$

which expresses this probability in terms of the likelihood of observing the data (given the model parameters) and the prior distribution of the parameters which represents our a priori knowledge about the model. In the context of neuronal activity, the latent variables $Z$ describe how neurons are organized into assemblies. Therefore, to perform statistical inference, we first need to introduce a model which allows us to calculate the likelihood of both observed neuronal activity and the latent configuration representing assembly structure.

### The generative model

In this section we outline our approach to characterize neuronal activity of $N$ neurons organized into $A$ assemblies for $M$ time frames. We assume that each neuron belongs to one assembly and denote their memberships by vector of integer labels $\{t_1, \cdots, t_N\}$ between 1 and $A$. The state of all assemblies over time is specified by a binary matrix $\omega$ (0 = "off", 1 = "on") of size

$M \times A$. In the following we will adopt the shorthand notation $\boldsymbol{\mu} = \{i \in \{1, \cdots, N\} : t_i = \mu\}$ to indicate the set of neurons assigned to the same assembly $\mu$. Vectors of parameters will be denoted by dropping their indexes, e.g. $p \equiv \{p_1, \cdots, p_A\}$. The neuronal memberships and the assembly activity matrix are unobserved variables that we want to estimate from the observed neuronal activity data.

Our model is specified by the following three generative steps: (1) draw neuronal membership $t_i$ from a categorical distribution with probabilities $n_\mu$, $\mu = 1, \ldots, A$; (2) draw independently the activity states $\omega_{k\mu} = \{0, 1\}$ of assembly $\mu$ at time $k$ from a Bernoulli distribution with assembly-specific probabilities $p_\mu \equiv P(\omega_{k\mu} = 1)$; (3) draw the activity of all neurons at all times, denoted by the binary matrix $s_{ik}$, from the conditional probability

$$\lambda_{t_i}(z) \equiv P(s_{ik} = 1 | \omega_{kt_i} = z) \tag{2}$$

which depends on the state $z \in \{0, 1\}$ of the corresponding assembly $t_i$. The parameters $\lambda_\mu(1)$ and $\lambda_\mu(0)$ represent the probabilities of any of the neurons belonging to the assembly $\mu$ to fire when the assembly is active or inactive respectively. From here onward we will refer to $\lambda_\mu(0)$ as the level of asynchrony in the assembly and to $\lambda_\mu(1)$ as assembly synchrony, characterizing the propensity of the assembly's constituent neurons to fire synchronously.

The model parameters $\theta = \{n, p, \lambda\}$ provide a full characterization of the assemblies based on their statistical properties, including firing frequency, size, synchrony and asynchrony levels. As discussed in the following sections, our approach allows us to estimate the model parameters from the data together with the latent variables of the model.

## Likelihood

From the generative rules outlined in the section above, we can calculate the joint probability of neuronal membership $t$, the assembly activity matrix $\omega$ and the neuronal activity matrix $s$ conditional to the model parameters $\theta$ (likelihood) as

$$P(t, \omega, s | \theta) = \left( \prod_{i=1}^{N} n_{t_i} \right) \cdot \left( \prod_{\mu=1}^{A} \prod_{k=1}^{M} p_\mu^{\omega_{k\mu}} (1 - p_\mu)^{1 - \omega_{k\mu}} \right) \cdot$$
$$\cdot \left( \prod_{i=1}^{N} \prod_{k=1}^{M} [\lambda_{t_i}(\omega_{kt_i})]^{s_{ik}} [1 - \lambda_{t_i}(\omega_{kt_i})]^{(1 - s_{ik})} \right) \tag{3}$$

Next, we need to define the distributions used as priors on the model parameters, as indicated in Eq (1). In particular, we use a beta distribution for the assembly activation probabilities $p$'s and the synchronous/asynchronous firing probabilities $\lambda$'s whereas for the relative sizes of each assembly $n$'s we employ a Dirichlet distribution which imposes the normalization condition $\sum_\mu n_\mu = 1$. Our prior distributions are then summarized as

$$p_\mu \sim \text{Beta}(\alpha_\mu^{(p)}, \beta_\mu^{(p)}), \tag{4}$$

$$\lambda_\mu(z) \sim \text{Beta}(\alpha_{z,\mu}^{(\lambda)}, \beta_{z,\mu}^{(\lambda)}), \tag{5}$$

$$\{n_1, \cdots, n_A\} \sim \text{Dir}(\alpha_1^{(n)}, \cdots, \alpha_A^{(n)}) \tag{6}$$

where the standard notation $x \sim P$ means that $x$ is drawn from the distribution $P$. $\alpha$'s and $\beta$'s are the (hyper-)parameters describing our prior knowledge on the model parameters such as the expected temporal sparsity of the synchronous activation of neuronal populations. This model can be represented graphically as in S1 Fig (panel A).

The present formulation of the generative model could be used directly to draw posterior samples of parameters and latent features by deriving their full conditional distributions (Gibbs sampling). However, this strategy would introduce several problems in dealing with a variable number of assemblies, as continuous features (such as the vector $n$ of relative size for instance) would have variable dimensionality. We can derive a more efficient sampler by integrating out the continuous parameters $\theta$ and obtaining the marginal probability $P(t, \omega, s)$. This procedure leads to the "collapsed" model depicted in S1 Fig (panel B) which in general reduces the uncertainty associated with the estimation of the remaining variables. To proceed our derivation we first introduce the summary variables

$$G_\mu = \sum_{i=1}^{N} \delta_{\mu, t_i} \tag{7}$$

$$H_\mu = \alpha^{(p)} + \sum_{k=1}^{M} \omega_{k\mu}, \quad \bar{H}_\mu = \beta^{(p)} + \sum_{k=1}^{M} (1 - \omega_{k\mu}) \tag{8}$$

$$\hat{T}_{S;\mu k}^{zz'} = \sum_{i \in S} \delta_{z, \omega_{k\mu}} \delta_{z', s_{ik}}, \quad T_{S;\mu}^{zz'} = \gamma_{zz'} + \sum_{k=1}^{M} \hat{T}_{S;\mu k}^{zz'} \tag{9}$$

$$\gamma_{z1} = \alpha_z^{(\lambda)}, \quad \gamma_{z0} = \beta_z^{(\lambda)} \tag{10}$$

where $\delta_{ij}$ is the Kronecker delta function and $S$ is any subset of indexes in the range 1 to $N$. To simplify the notation we also introduce additional matrices derived from Eq (9)

$$\hat{T}_{\mu k} = \hat{T}_{\boldsymbol{\mu};\mu k}, \quad T_\mu = T_{\boldsymbol{\mu};\mu} \tag{11}$$

$$\hat{T}_{\mu \setminus i;k} = \hat{T}_{\boldsymbol{\mu} \setminus i;\mu k}, \quad T_{\mu \setminus i} = T_{\boldsymbol{\mu} \setminus i;\mu}, \tag{12}$$

$$\hat{T}_{i;\mu k} = \hat{T}_{\{i\},\mu k}, \quad T_{i,\mu} = T_{\{i\},\mu} \tag{13}$$

For each assembly $\mu$, $G_\mu$ is the assembly size, $H_\mu$ and $\bar{H}_\mu$ denote (up to an additive constant) the number of active and inactive events over time respectively, while the matrix $T_{S,\mu}^{zz'}$ counts how many times the state $(\omega_{k\mu}, s_{ik}) = (z, z')$ is observed across time and the neurons in the set $S$ (up to an additive constant).

Due to the conjugate character of the prior distributions (see Materials and methods for details) the integration over $\theta$ can be carried out analytically, leading to the marginal likelihood

$$P(t, \omega, s) = \int d\theta \; P(t, \omega, s | \theta) P(\theta) =$$

$$= \frac{\mathcal{B}(G + \alpha^{(n)})}{\mathcal{B}(\alpha^{(n)})} \prod_{\mu=1}^{A} \left\{ \frac{B(H_\mu, \bar{H}_\mu)}{B(\alpha^{(p)}, \beta^{(p)})} \cdot \prod_{z \in \{0,1\}} \frac{B(T_\mu^{z1}, T_\mu^{z0})}{B(\alpha_z^{(\lambda)}, \beta_z^{(\lambda)})} \right\} \tag{14}$$

where $B(\cdot, \cdot)$ is the Euler beta function and $\mathcal{B}$ is defined as the product of gamma functions

$$\mathcal{B}(\{\alpha_1, \cdots, \alpha_n\}) \equiv \frac{\prod_k \Gamma(\alpha_k)}{\Gamma(\sum_k \alpha_k)}. \tag{15}$$

After integrating out $\theta$ we can rewrite the joint probability in Eq (14) as the product between

the likelihood

$$P(t, s|\omega) = \frac{\mathcal{B}(G + \alpha^{(n)})}{\mathcal{B}(\alpha^{(n)})} \cdot \prod_{\mu=1}^{A} \prod_{z \in \{0,1\}} \frac{B(T_\mu^{z1}, T_\mu^{z0})}{B(\alpha_z^{(\lambda)}, \beta_z^{(\lambda)})} \tag{16}$$

and the prior on the latent variables

$$P(\omega) = \prod_{\mu=1}^{A} \left\{ \frac{B(H_\mu, \bar{H}_\mu)}{B(\alpha^{(p)}, \beta^{(p)})} \right\}. \tag{17}$$

We will employ this "collapsed" formulation of the model to make inference on cell membership and assembly activity.

## Inference

Given the data on neural activities in the form of a binary matrix $s$, we can use the generative model described above to make inference on neuronal identities and assembly activity along with the model parameters. Within our probabilistic framework, we can estimate these quantities by evaluating their expectations with respect to the conditional distribution $P(t, \omega|s)$. These averages can be computed via Monte Carlo methods by generating random samples from the joint distribution.

We first consider the case where the number of neuronal assemblies $A$ is known. To obtain samples from $P(t, \omega, s)$ in Eq (14) we can use a Gibbs sampler where cell membership and assembly activities are drawn sequentially from the conditional distributions $P(t|\omega, s)$ and $P(\omega|t, s)$ respectively (Algorithm 1).

**Algorithm 1** Collapsed Gibbs sampling
```
1: Initialize t
2: while convergence criteria do
3:    for each assembly μ ∈ {1, ···, A} time k ∈ {1, ···, M} do
4:        draw ω_{kμ} ∼ P(ω_{kμ}|t, s)
5:    for each cell i ∈ {1, ···, N} do
6:        draw t_i ∼ P(t|ω, s)
7:    draw θ ∼ P(θ|t, ω, s)
```

The conditional distribution of the membership of the $i$th data point $t_i$ can be written as

$$P(t_i = \mu|t_{-i}, \omega, s) \propto (G_\mu + \alpha_\mu^{(n)}) \prod_{z \in \{0,1\}} \frac{B(T_\mu^{z1}, T_\mu^{z0})}{B(T_{\mu \backslash i}^{z1}, T_{\mu \backslash i}^{z0})} \tag{18}$$

where we used the notation $t_{-j} = \{t_1, \cdots, t_{j-1}, t_{j+1}, \cdots, t_N\}$.

The conditional probability for the assembly activation matrix $\omega$ is given by

$$P(\omega_{kμ} = 1|t, \omega_{-kμ}, s) = \frac{1}{1 + \rho_{kμ}} \tag{19}$$

$$\rho_{kμ} = \frac{H_\mu + \alpha^{(p)}}{\bar{H}_\mu + \beta^{(p)}} \cdot \prod_z \frac{B(T_\mu^{z1}, T_\mu^{z0})|_{\omega_{kμ}=0}}{B(T_\mu^{z1}, T_\mu^{z0})|_{\omega_{kμ}=1}}, \tag{20}$$

sampling the parameters $\theta$ from the posterior distribution $P(\theta|t, \omega, s)$ is easy thanks to our choice of the corresponding priors. The posterior distributions of $p_\mu$, $\lambda_\mu$ and $n_\mu$ retain the same

functional form as in Eqs (4, 5 and 6) with updated hyper-parameters

$$\tilde{\alpha}_\mu^{(p)} = H_\mu, \quad \tilde{\beta}_\mu^{(p)} = \bar{H}_\mu \tag{21}$$

$$\tilde{\alpha}_{0,\mu}^{(\lambda)} = T_\mu^{01}, \quad \tilde{\beta}_{0,\mu}^{(\lambda)} = T_\mu^{00} \tag{22}$$

$$\tilde{\alpha}_{1,\mu}^{(\lambda)} = T_\mu^{11}, \quad \tilde{\beta}_{1,\mu}^{(\lambda)} = T_\mu^{10} \tag{23}$$

$$\tilde{\alpha}_\mu^{(n)} = \alpha_\mu^{(n)} + G_\mu \tag{24}$$

where we have added an extra label $\mu$ to the posterior hyperparameters corresponding to different assemblies. Having obtained the analytical expression of the marginal likelihood for the collapsed model in Eq (14) we have designed a Gibbs Monte Carlo sampler which allows us to make inference on latent features and model parameters. However, so far we have kept the number of assemblies $A$ fixed. In the next section we will discuss how to extend the current sampling strategy to accommodate a variable number of assemblies, providing a way to estimate this number directly from the data.

## Inference of the number of neuronal assemblies

Since in general the number of assemblies is not known a priori, we need to extend the framework described in the previous section in order to estimate this number from the data. This can be achieved by introducing a scheme in which the number of assemblies is treated as a latent feature for which we need to draw posterior samples analogously to the other parameters of the model.

Estimating the number of components in a mixture is a common issue in unsupervised machine learning. In the context of our model we employ the Dirichlet process(DP) [20] as a prior on the mixing measure describing the composition of the data (see panel C in S1 Fig for a graphical representation). The DP prior can be implemented by introducing specific Metropolis-Hastings acceptance rules to increase or decrease the number of components [21], which generalize the inference problem to an arbitrary number of assemblies.

Here we adapted the algorithm described in Neal [21] to our generative model of neuronal assemblies. In particular, during each iteration of the sampler, when assigning a given neuron $i$ to an assembly, the rule proposed by Neal corresponds to draw a new membership $\mu^*$ from the proposal

$$q(\mu^*) = \begin{cases} \frac{G_{\mu^*}^{(-i)}}{\alpha+N-1} & \mu^* = 1, \cdots, A \\ \frac{\alpha}{\alpha+N-1} & \mu^* = A+1 \end{cases} \tag{25}$$

where $\alpha$ is the concentration parameter of the Dirichlet process. Existing assemblies are drawn proportionally to their occupancy $G_\mu^{(-i)}$ (calculated from all membership except the current $\mu_0$, see Ref. [21] for details). If a new assembly is proposed, $\mu = A+1$, then we draw a corresponding binary vector $\omega_{k\mu}$ from the prior distribution in Eq (17). The proposal is accepted with probability

$$a(\mu^*, \mu_0) = \min\left\{1, \prod_{z \in \{0,1\}} \frac{[B(T_{\mu^*}^{z1}, T_{\mu^*}^{z0})B(T_{\mu_0}^{z1}, T_{\mu_0}^{z0})]_{t_i=\mu^*}}{[B(T_{\mu^*}^{z1}, T_{\mu^*}^{z0})B(T_{\mu_0}^{z1}, T_{\mu_0}^{z0})]_{t_i=\mu_0}}\right\}. \tag{26}$$

With this Metropolis-Hastings rule we have introduced transitions in the number of neuronal assemblies. By integrating this rule to the collapsed Gibbs sampler we obtain a new sampling scheme (Algorithm 2) which allows us to infer the number of neuronal assemblies from the data.

**Algorithm 2** Metropolis-Hastings sampling with Dirichlet process prior

```
Initialize t
while convergence criteria do
  for each assembly μ ∈ {1, ⋯, A} time k ∈ {1, ⋯, M} do
    draw ω_{kμ} ∼ P(ω_{kμ}|t, s)
  for each cell i ∈ {1, ⋯, N} do
    draw μ* ∼ q(μ)
    if μ* = A + 1 then
      draw ω_{kμ*} ∼ P(ω)
      A → A + 1
     Accept new μ* with probability a(μ*, μ_0)
    if G_{μ_0} = 0 then
      delete assembly μ_0
      A → A - 1
  draw θ ∼ P(θ|t, ω, s)
```

## Validation

To validate the method we generated random neuronal activity matrices $s$ from our model by fixing synchrony, asynchrony and activity for each neuronal assembly, then we used the Monte Carlo algorithm 2 outlined in the previous section to infer parameters and latent variables. In Fig 1 we illustrate a sample of the activity matrix $s$ obtained from the model with five assemblies with cells sorted by membership (Fig 1A). At the initialization step we assigned random membership to each cell uniformly between 1 to $N$, generating an initial number of assemblies of order $N$. In Fig 1C we display the course of the reassignment for each neuron at four different stages of the sampling. In particular, the early phase where initial groups are formed is followed by the merging of equivalent assemblies which eventually converge to the true assigned membership. Fig 1B displays the convergence of the log-likelihood (top panel) and the membership transition rate, defined as the number of cells reassigned to a different assembly divided by the total number of cells (bottom panel). When the data display a strong organization in terms of synchronous activity, deviations from the maximum likelihood configuration are suppressed. Therefore once cells are assigned to their ground truth label (up to permutation) they are unlikely to be reassigned after the sampler has reached convergence.

Next, we explored a broad range of synchrony and asynchrony to describe the regime of applicability of our method. We performed inference on random activity matrices generated from the model with asynchrony and synchrony in the range between 0.05 and 0.95. The diagram depicted in Fig 2A shows the separation between regions where all cell memberships are exactly recovered (detectable phase, red) from regions where full recovery of cell membership is not achievable (non-detectable phase, blue). In particular, assembly recovery is possible whenever the absolute difference between asynchrony and synchrony $|\lambda(1) - \lambda(0)|$ is sufficiently high. Indeed, the condition $\lambda(1) = \lambda(0)$ corresponds to the degenerate case where the firing probability is independent of the assembly state, in which case $s$ does not provide any statistical information about cell membership. The width of the non-detectable region depends on assembly activity $p_\mu$'s; at fixed time $M$ higher assembly activity provides more information. Therefore, when the activity increases the size of the non-detectable region is reduced (S2 Fig). The detectable phase is separated in two symmetric regions. $\lambda(1) > \lambda(0)$ corresponds to "on" assemblies where neuron activity is correlated with assembly activity. The detectable region

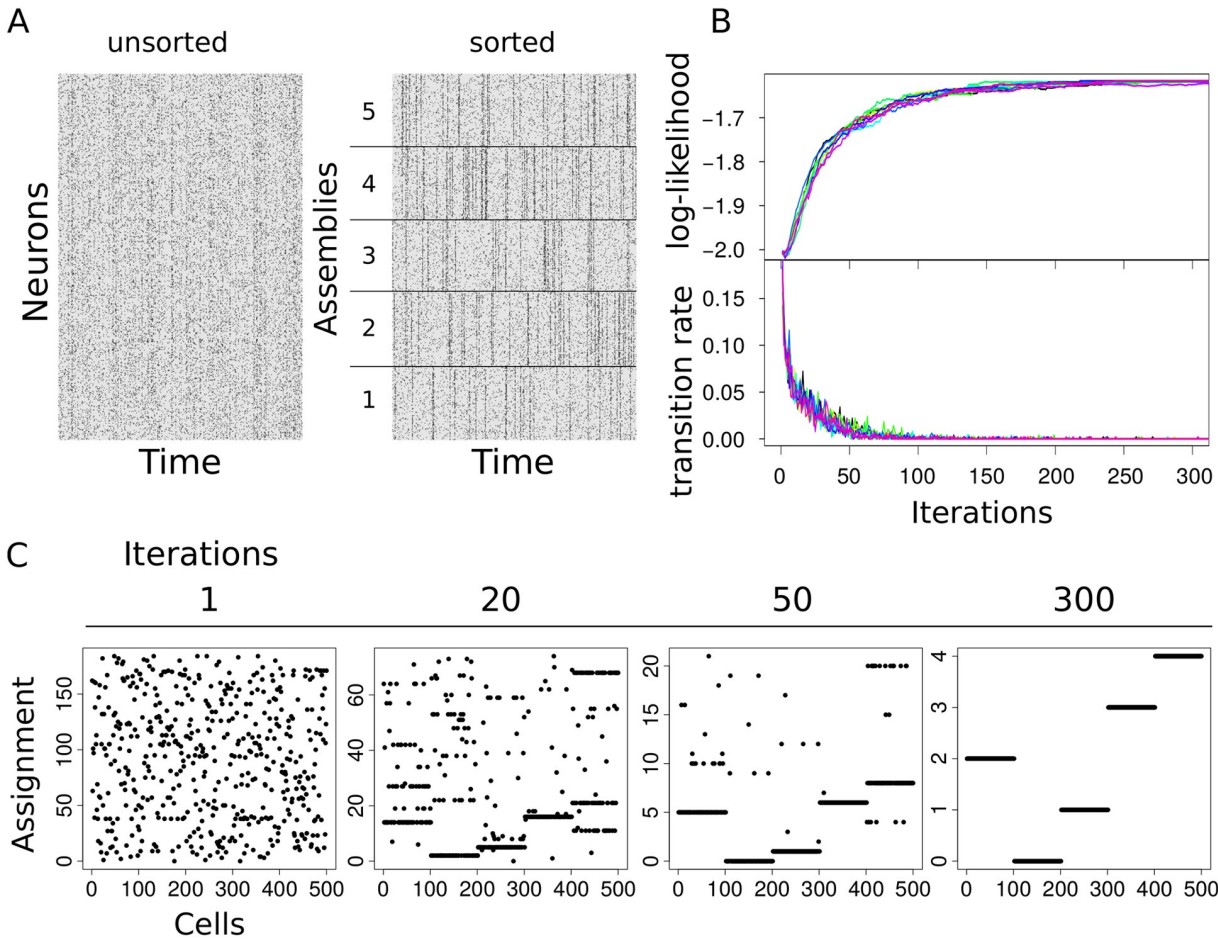

**Fig 1. Validation.** (A) Simulated activity of $N = 500$ neurons organized in $A = 5$ assemblies. Time sequences of length $M = 1000$ for all neurons have been generated from the model with $\lambda_\mu(0) = 0.08$, $\lambda_\mu(1) = 0.6$ and $p_\mu = 0.1$ for asynchrony, synchrony and activity equal for all assemblies. We used this activity matrix as input to our inference algorithm to recover the five assemblies. (B) (top) log-likelihood over the course of 300 iterations of the Monte Carlo sampler and (bottom) number of transitions per iteration divided by the number of neurons $N$ (different colors correspond to different initializations of neuronal memberships). (C) Assignments of cells to assemblies at four stages of the sampling. Cells initially assigned to $\mathcal{O}(N)$ assemblies are progressively grouped until the original assemblies are recovered.

with $\lambda(1) < \lambda(0)$ corresponds to "off" assemblies where member neurons are anti-correlated with assembly activity. This regime describes the scenario where a population of highly active neurons is characterized by synchronous transitions to "off states". The average membership transition rate can be viewed as an order parameter for the detectable/non-detectable phase transition. In fact, unlike the detectable phase where the membership transition rate vanishes, in the non-detectable phase this quantity reaches a non-zero limit, as shown in Fig 2B, indicating that maximum-likelihood fluctuations are no longer suppressed. In this regime, the information provided by the activity matrix $s$ is not sufficient to assign neurons to an assembly.

## Comparison with k-means, dimensionality reduction and spectral clustering methods

In this section we compare the performance our technique with other methods widely used for clustering. The data used to test the performance of the various methods were simulated according to a different generative model from the one used for inference. This

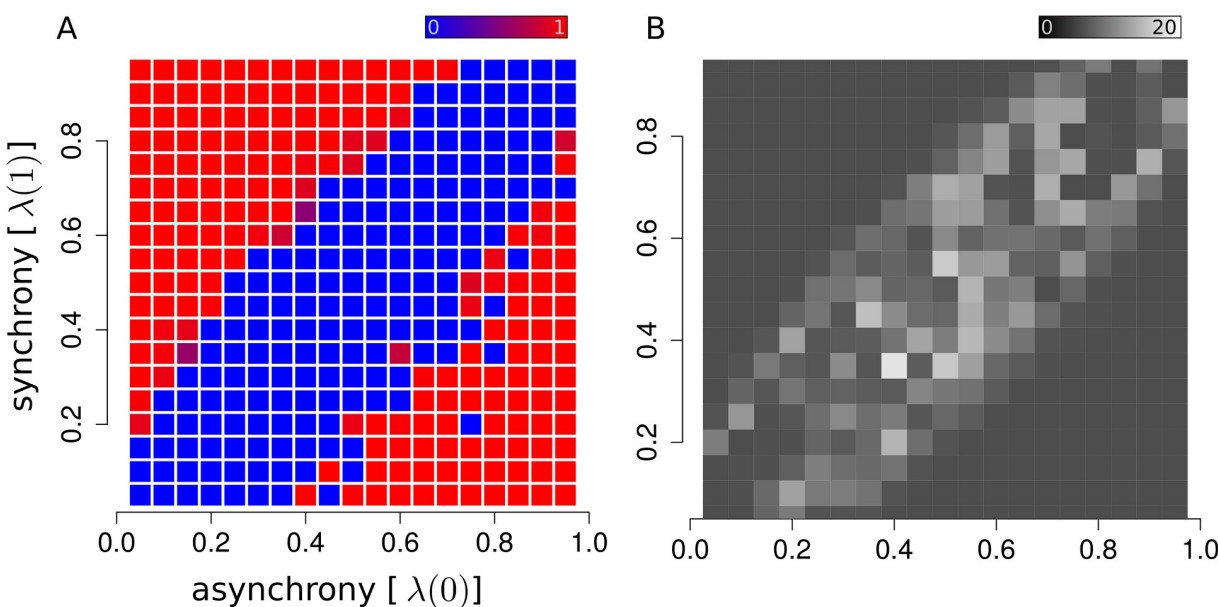

**Fig 2. Phase diagram.** The ability to identify assemblies in a dataset depends on the model parameters. When synchrony and asynchrony levels are too similar, the recording time might not be sufficient to capture small differences in firing patterns. (A) Raster plot displaying detectable (red) versus non-detectable (blue) regimes of as a function of synchrony and asynchrony parameters of the model (we used 5 randomly generated assemblies for each configuration of synchrony and asynchrony). (B) Raster plot representing the number of neurons reassigned, on average, for every random sample of the Markov chain. The average transition rate can be viewed as an order parameter characterizing the transition between detectable and non-detectable phases.

was done to deviate from the optimal scenario where testing data are generated from the same model used for inference (which would necessarily make our method outperform the others). Specifically, 20% of all the neurons were allowed to belong to multiple assemblies ($< 3$) and can be recruited by any assembly they are part of according to synchrony and asynchrony levels in each assembly (i.e. the probabilities $\lambda(z)$ to be active conditional to the assembly state $z$). To compare the membership assignments obtained from each method we defined a measure of performance as follows. For any assignment map $i \rightarrow \tau(i)$ defining the membership for all neurons $i = \{1, \cdots, N\}$ we can construct the pairwise assignment matrices $I_{ij}^{(\tau)}$ indicating whether the pair of neurons $i, j$ have been assigned to the same assembly ($I_{ij}^{(\tau)} = 1$) or not ($I_{ij}^{(\tau)} = -1$). A measure of performance can be obtained as a correlation between the original pairwise assignment matrix and the one obtained from $\tau$ which can be written as

$$\rho_\tau = \frac{1}{K} \sum_{i<j} I_{ij}^{(\tau)} I_{ij}^{(orig)} \qquad (27)$$

where $I_{ij}^{(orig)}$ is the pairwise assignment matrix obtained from the ground-truth assignment (see Materials and methods for details on the generalization to the case of multiple membership). The number of pairs $K = N(N - 1)/2$ is used for normalization. We compared the performance of our method to k-means, PCA and spectral clustering (see Materials and methods for details). We analyzed simulated neuronal activity for 1000 time frames at increasing levels of asynchrony, number of neurons and assemblies (Fig 3A–3C). In all comparisons k-means was instructed with a number of assemblies estimated with the silhouette method. For the PCA-based method we selected the number of significant components by comparing with circularly

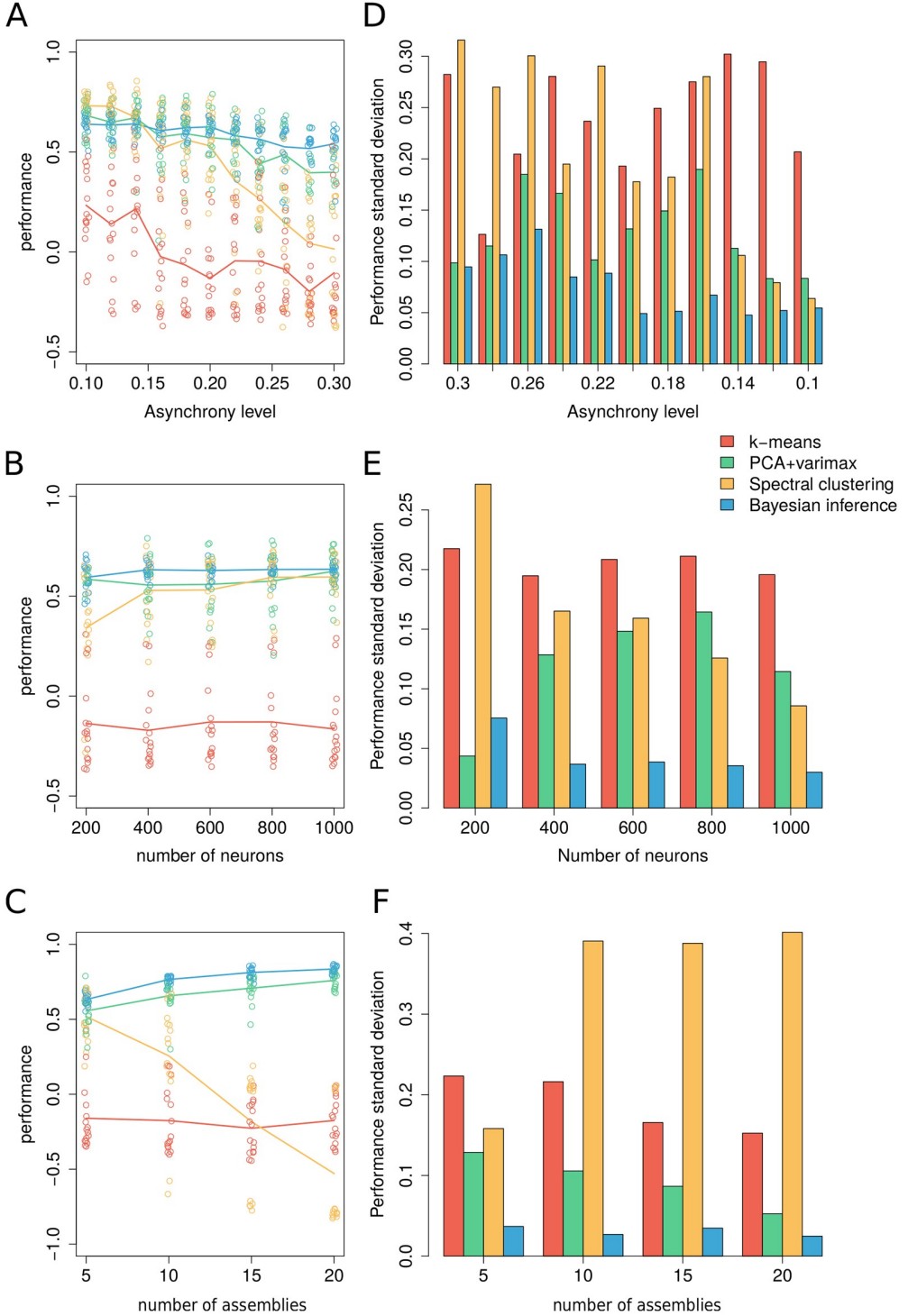

**Fig 3. Comparison of Bayesian inference performance to *k*-means, PCA and spectral clustering.** (A) Performance comparison across levels of asynchrony. Dots correspond to independently generated data sets while solid lines show the average performance for each method over all simulated data. (B) Comparison across number of neurons. (C) Comparison across number of assemblies. (D-F) Standard deviation of the performance across simulated data per parametric condition. Unless specified otherwise, surrogate datasets were generated using 400 neurons and 1000 time frames distributed over 5 assemblies with assembly activity of 5%, synchrony 50% and asynchrony 10%. For *k*-means we used a number of clusters estimated according to the silhouette method. The number of significant principal components in PCA clustering was obtained by the circular shuffling method whereas for the spectral clustering we used the Newman-Reinart graph-theoretic community detection method [22] (see Materials and methods). The performance was measured according to Eq (27) by comparing each set of assignments with ground truth assignments.

shuffled data as a null model, while for spectral clustering we used a community detection algorithm to estimate the number of assemblies (see Materials and methods for details). As reported in previous studies, k-means is not suitable for detecting neuronal assemblies as shown by the low performance in all conditions due to the high dimensionality of the data [16]. Such low performance stems from the difficulty in detecting the right number of assemblies. PCA and spectral clustering have good performance for asynchrony levels below 20%. For the specific testing datasets used in our comparisons, below 15% asynchrony and low number of assemblies, the spectral method can cope better with multi-membership, leading to higher performance. The spectral clustering method hinges on the graph representation of the neuronal activity (see Materials and methods). Depending on the type of data, this representation can lead to systematic under- or over-estimation of the number of assemblies, requiring additional procedures [5, 16] in the pipeline. In a broad range of model parameters, Bayesian inference outperforms all the other methods. Next we considered performance variability across simulated data. In the range of parameters analyzed, Bayesian inference displayed a minimal variability compared to other methods, showing that our technique provides a very reliable inference of neuronal membership and it is robust across asynchrony, number of neurons and assemblies (Fig 3D–F).

In general, performance of different clustering methods depend on the generative rules used to simulate the testing data. For example, the performance of all methods tested here improves when neurons belong to only one ensemble (see S8 Fig). Therefore, when analysing real neuronal data, it is important to be aware of the hypotheses underlying the methods used for clustering. Our method is specifically tailored for neuronal recordings and it makes minimal assumptions on how neuronal activity is hierarchically structured into synchronous and asynchronous assembly activity and uses these as defining features.

## Analysis of assemblies in the zebrafish tectum

We used our method to infer the structure and dynamics of neural assemblies within the optic tectum of the larval zebrafish. For these experiments we used zebrafish with pan-neuronal expression of a nuclear localized calcium indicator, Tg(*elavl3:H2B-GCaMP6s*) [23] (see Materials and methods). Using two-photon volumetric calcium imaging, we monitored activity in the tectum of immobilized zebrafish larvae kept in total darkness. In all experiments we recorded calcium activity for 1 hour throughout 5 planes, $15\mu$m apart with an acquisition frequency of 4.8Hz per volume (Fig 4A, S1 Video and Materials and methods). Volumetric images were preprocessed using registration and segmentation software to extract the time sequences from thousands of tectal cells in both tectal hemispheres (Fig 4B and Supplementary Information). Consistent with previous reports we find that the tectum displays significant levels activity in the absence of visual stimulation [5, 6])

In order to match the data with our generative model we extracted the onset times of each calcium transient from all tectal cells in the form of a binary matrix (Fig 4C). To do so we separated calcium transient signals from baseline activity by using a hidden Markov model (HMM) where the neuronal activity (hidden) state $s_t$ at time $t$ is represented by a Bernoulli process, the background $b_t$ is a Gaussian Markov process and the calcium transients are drawn from a normal distribution when $s_t = 1$ or follow a deterministic exponential decay when $s_t = 0$ (Fig 4D and 4E). More realistic models have been developed to describe the fluorescence levels of the calcium indicator [24, 25]. However, since our purpose is to capture the synchronous events among cells and not to infer the number of spikes within a calcium transient we can use a simplified model to select events where the activity deviates significantly from the

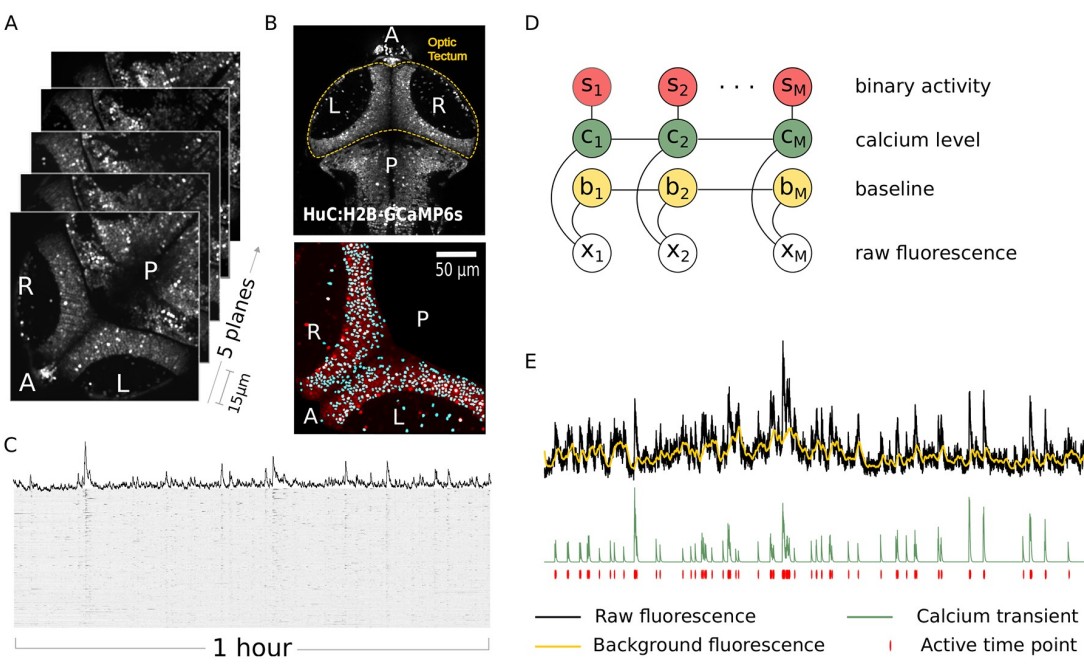

**Fig 4. Imaging of the zebrafish tectum.** (A) Volumetric 2-photon imaging of both tectal hemispheres showing anterior/posterior (A/P) and right/left (R/L) axis of the zebrafish optic tectum. We monitored activity in the tectum of immobilized zebrafish larvae kept in total darkness. In all experiments we recorded calcium activity for 1 hour throughout 5 planes, $15\mu$m apart with an acquisition frequency of 4.8Hz per volume. (B) Raw images (top) were segmented (bottom) to obtain the temporal dynamics of calcium of thousands of neuron in the tectum (see also S1 Video). (C) Raster plot showing calcium activity of all neurons over 1h recording. (D) Raw fluorescence is described in terms of a Hidden Markov model (HMM) where periods of increased activity are indicated by a binary process ($\{s\}$) triggering the onset of calcium transients ($\{c\}$). The recorded fluorescence is obtained as the sum of calcium level and a Wiener process to account for the low-frequency baseline modulation ($\{s\}$) (see Materials and methods). (E) Example of raw fluorescence and estimation of hidden variables of the HMM using a sequential maximum-likelihood algorithm (see Materials and methods).

background. The HMM approach can be used to obtain estimates of signal and background (see Materials and methods) allowing for fast preprocessing of the calcium traces.

Estimates of the binary activity for each neuron were then combined into matrices of dimension [$N$ cells] × [$M$ time frames] where $M$ = 17460 frames for all experiments and a number of neurons $N$ of order $10^3$ dependent on how many cells were segmented for each fish. To build the activity matrix $s$ used for inference we considered all time frames with more than 15 active cells. This threshold was used to select for synchronous activity events that had low probability ($p < 0.02$) of being random, obtained by shuffling neuronal activity independently over time. We applied our method to each binary activity matrix by running Algorithm 2 until convergence. The advantage of our method is that assembly features (activity, levels of asynchronous and synchronous activity) are simultaneously estimated together with neuronal membership. This allows the user to select particular assemblies for further analysis. Here, we have discarded assemblies composed of cells which were almost never active during the 1 hour recording period. Similarly, groups of poorly correlated neurons firing asynchronously can be combined into assemblies where levels of asynchrony and synchrony are very similar and therefore potentially outside the detectable regime. This allows us to automatically separate neurons which do not belong to coherent assemblies from neurons where the level of synchrony is significantly larger than the level of asynchrony (S6 Fig shows that neurons not assigned to an assembly 'free neurons' are only weakly correlated with assembly activity). To focus on the most coherent populations we selected assemblies with activity larger than 0.5%

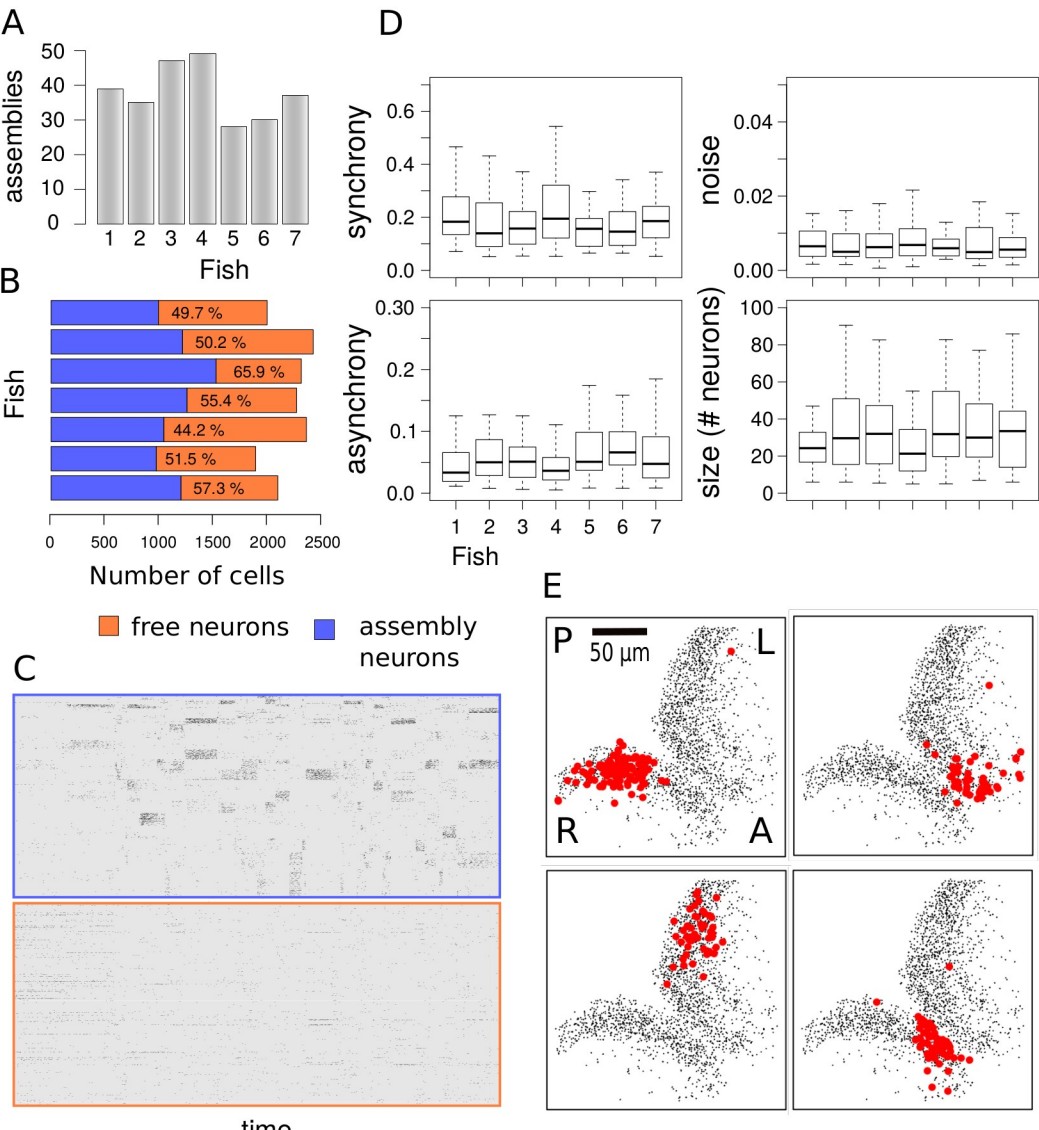

**Fig 5. Stereotyped features of neuronal assemblies across fish.** (A) Number of neuronal assemblies estimated for each fish. (B) Fraction of neurons assigned to assemblies with 99% confidence. The efficacy of our method in capturing neuronal assemblies is illustrated by separating the activity matrix of assembly neurons from neurons which are not part of a coherent assembly. (C) When sorting the activity matrix by membership, neurons that are part of an assembly generate bands of synchronous activity whereas neurons that are not assigned to an assembly display independent random firing events. (D) Comparison of synchrony, asynchrony, activity and assembly sizes for all experiments. Synchrony and asynchrony correspond to the model parameters $\lambda(1)$ and $\lambda(0)$, as the probabilities of a neuron to fire with its assembly (synchronous activation) and independently of it (asynchronous activation). The activity corresponds to the probability of an assembly to be active at any time during the 1 hour recording period. (E) Representative assemblies from single fish with high synchrony and activity display spatial compactness within left or right tectal hemispheres. Labels A/P and R/L indicate anterior/posterior axis and right/left optic tectum respectively. Assembly maps are obtained by projecting in 2D the positions of all member neurons across the five imaging planes.

(corresponding to $\approx$1 event per minute), size larger than 5 neurons, synchrony larger than 5% and asynchrony lower than 5%.

By comparing the analysis of $n = 7$ fish using our technique, we obtained a number of neuronal assemblies (Fig 5A) which varied between 30 and 50 per larva. All assemblies features

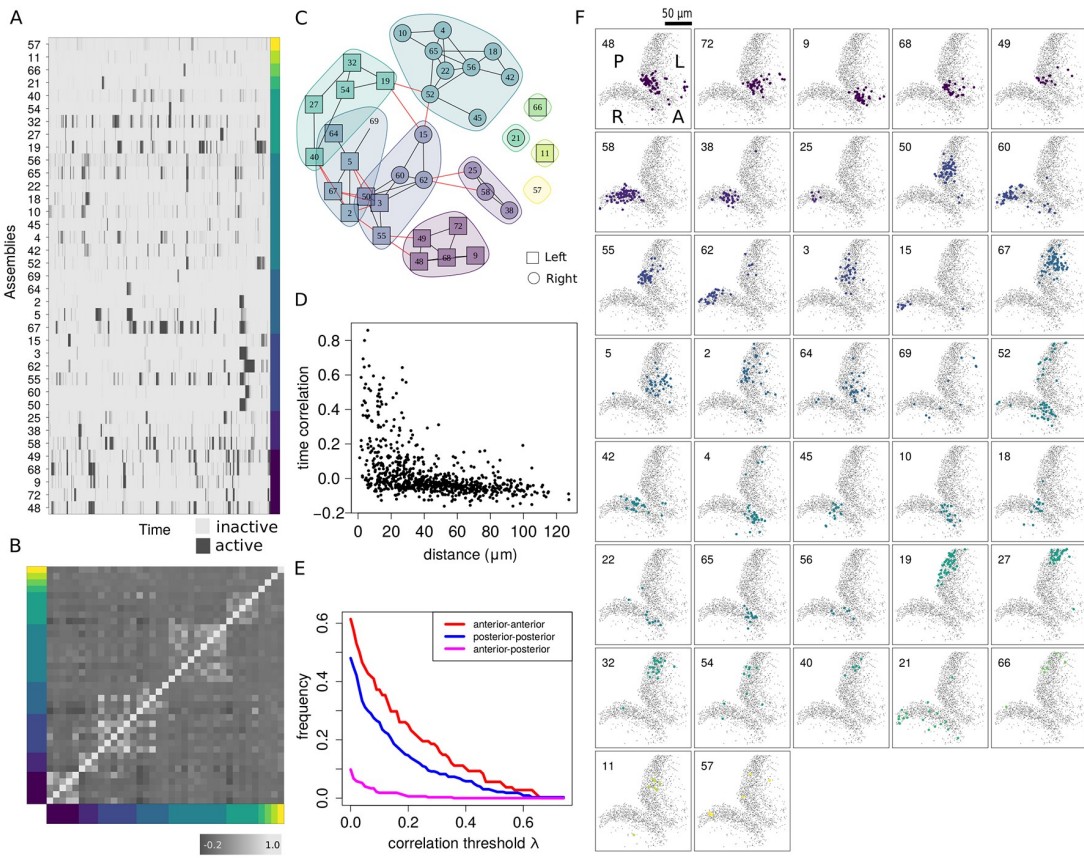

**Fig 6. Networks of assemblies in the zebrafish tectum.** (A) Raster plot of assembly activity ordered according to subnetwork membership as shown in C. (B) Raster plot of the time correlation matrix (Pearson's) between all pairs of assemblies averaged across posterior samples and sorted according to subnetwork membership. (C) Subnetwork graphs representing the time correlations among neuronal assemblies (see Materials and methods). Each assembly is a node, shorter edges reflect higher temporal correlations. Note the absence of edges between assemblies located in ipsilateral anterior and posterior tectum (see S3 Video). (D) Scatter plot showing the relationship between time correlation and physical distance between neuronal assembly centers (1pixel ≈ 0.9μm). (E) Fraction of assembly pairs with correlation larger than λ as a function of λ. Ipsilateral pairs of assemblies located at opposite sides in the anterior-posterior axis display a significant reduction in correlation compared to neighboring assembly pairs (anterior-anterior and posterior-posterior). (F) Locations of all assemblies from a single larva (see S2 Video for the assembly distributions in 3D). Each assembly is color-coded according to subnetwork memberships shown in C.

displayed non-significant variations across fish (Fig 5C). In particular, the average level of assembly synchrony across fish was 0.19 (SD = 0.11), corresponding on average to a 19% chance for a member neuron to be recruited during assembly activation. In addition, we have found typical asynchrony levels of 0.007 (SD = 0.005), meaning that there is a 0.7% chance of a neuron to be active at any time point where its assembly is off. The low level of synchrony is apparently low due to its definition in the model as a measure of coincident activity of member neurons at equal times (within the temporal resolution of the recording). However, assembly activity is characterized by sustained periods of activation during which most member neurons are recruited gradually rather than instantaneously.

Our inference method allows us to accommodate this feature as illustrated in Fig 6A where the assembly activity matrix displays extended periods of activity. Therefore, in spite of the local definition of synchrony in our model, windows of sustained activity emerge naturally as the optimal way to explain neuronal activity patterns in terms of assembly dynamics. This

property does not depend on how the raw neuronal activity is binarized. Instead, it is due to the hierarchical structure of our model.

Next we examined how the assemblies selected according to their features are distributed in the tectum. We found that over 90% of all detected assemblies were spatially compact within one of the two tectal hemispheres (Figs 5E and 6 and S7 Fig) and in their 3D distribution (see S2 Video). In some of the assemblies we observed the presence of a small number of "satellite" cells which were spatially separate from the assembly core and which were located either in the ipsilateral or contralateral tectum. Up to 66% of the active neurons were assigned to one of the selected assembly with 99% of confidence (Fig 5B), showing that neuronal assemblies recruit over a half of the tectal population over 1 hour of recording (Fig 5D).

The estimated assembly activity over time (Fig 6A) provides a coarse-grained description of the on-going activity in the optic tectum (see S3 Video). Although we did not assume any correlation among the assemblies, we can use the assembly time sequences inferred from the data to explore their relationships in the form of network graphs. The (Pearson's) time correlation between neuronal populations (Fig 6B, see also Supplementary Information) revealed the presence of subnetworks of assemblies (Fig 6C). These subnetworks are largely composed of neighboring ipsilateral assemblies with rarer edges between assemblies in opposite tectal hemispheres. This is shown in Fig 6C by representing the Pearson's correlation matrix between assembly time sequences as a graph with nodes corresponding to the neuronal assemblies and edges representing a positive correlation (with 95% confidence). Shorter edges reflect higher time correlation. Inspection of these graphs reveals that edges between assemblies located in anterior and posterior regions of a tectal hemisphere are almost never observed, as shown by the decrease of time correlation at large spatial separation between ipsilateral pairs of assemblies (Fig 6D and 6E). These results suggest functional segregation of anterior and posterior tectal networks. Moreover, the degree of correlation between assemblies located in opposite tectal hemispheres (even at a similar topographic location) is generally lower than the correlation between ipsilateral assemblies.

### Analysis of functional imaging data from the mouse cortex

To illustrate the general applicability of our method we also carried out analysis of neuronal assemblies in the mouse cortex (Fig 7A). We analyzed a dataset generated by Stringer et al. [19, 26] where the activity of over 10,000 neurons in the mouse visual cortex were recorded in head-fixed mice that were free to run on a air-floating ball. As we did for the zebrafish tectal data, we first applied our HMM technique (see Materials and methods) to extract calcium transients from the raw fluorescence of each recorded neuron to obtain the binarized neuronal activity matrix. Then we applied our Bayesian model to detect neuronal assemblies from the binary activity and infer their properties. Fig 7B displays the assembly activity over time for all assemblies satisfying the same selection criteria of activity, size, synchrony and asynchrony used for the zebrafish tectum. The majority of the cortical assemblies are distributed across the volume imaged (see S4 Video representing assembly 24). Some assemblies however, are organized into columns spanning through the z-axis (S5 Video, assembly 43) or are limited to ventral or dorsal planes (S6 Video, assembly 23). We analyzed the time correlation between assembly activities obtained with our method and some of the behavioral features (running speed and pupil area) which were simultaneously monitored during functional imaging in Ref. [19]. Several assemblies are characterized by a remarkably positive or negative correlation with running speed, consistent with the finding in Ref. [19] that the first principal component of the neuronal activity is highly correlated with arousal. In particular, the activity of the most populated assembly (assembly 24, Fig 7C) is anti-correlated with running speed, as opposed to

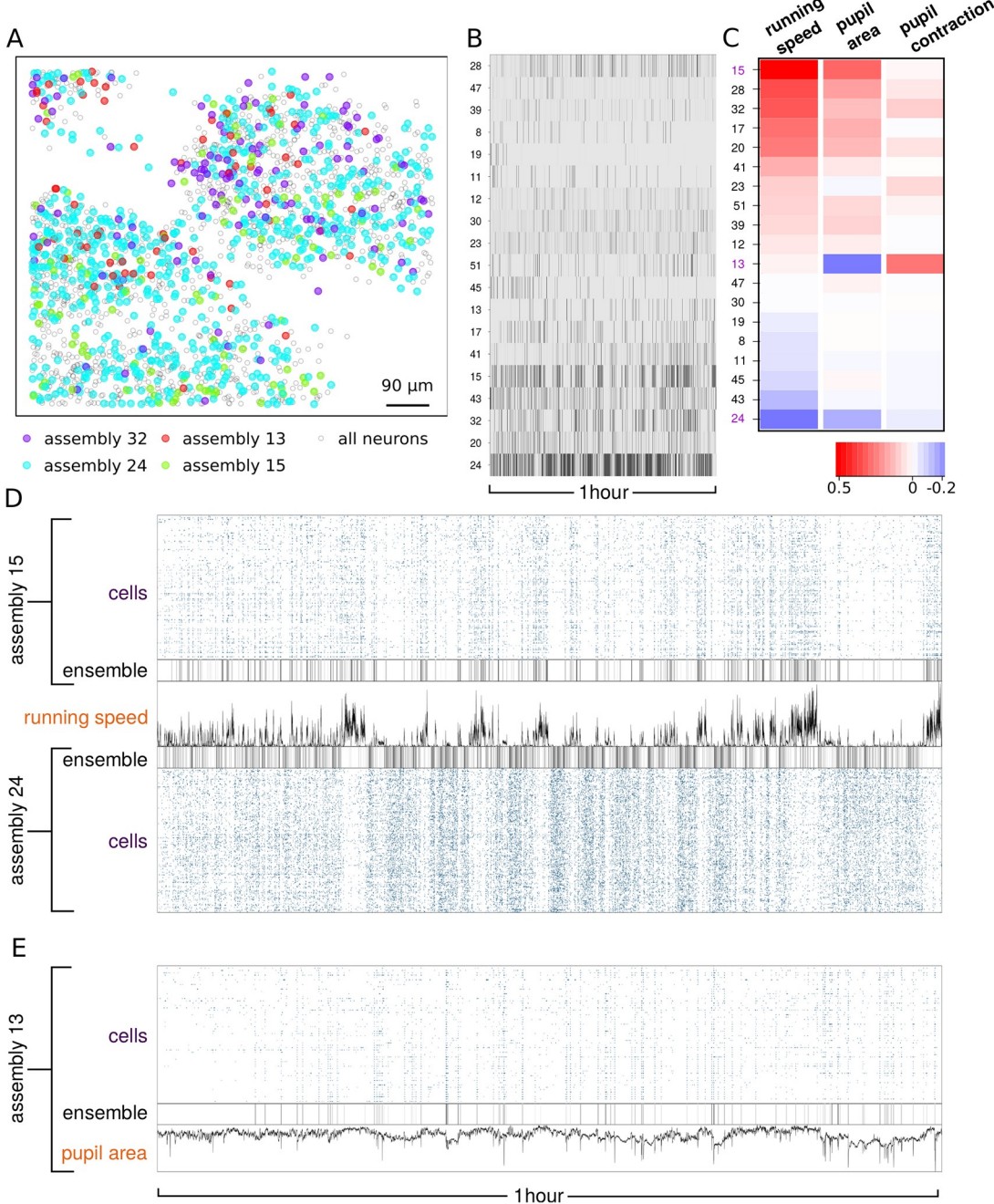

**Fig 7. Neuronal assemblies from functional imaging of the mouse cortex.** (A) 2D spatial distribution of neurons recorded from the mouse visual cortex from Ref. [19]. The color code represents selected assemblies obtained with our method. (B) Assembly activity matrix. (C) Correlation between the activity of each assembly and (from left to right) the time course of running speed, pupil area and the (binarized) pupil contraction. (D) Correlation and anti-correlation of neuronal assemblies with respect to running speed. (E) Assembly correlated with pupil contraction.

assembly 15 shown in Fig 7C and 7D which tends to be active during periods of running. Assemblies can be correlated with running speed at different time points or differ in activity and synchrony (see S5 Fig for a comparison between assemblies 15 and 32), suggesting that arousal is associated with the coordinated activation of multiple assemblies. By using the time

course of the pupil area we constructed a binary vector indicating the events of pronounced pupil contraction. One of the detected assemblies (assembly 13, Fig 7C) is highly correlated with pupil contraction events, as shown in Fig 7E. The striking correlation or anticorrelation of assembly activation with various behaviors suggest that our method is detecting assemblies that are behaviorally-relevant.

### Identification of assemblies in neuropixels recordings in mouse

Neuropixels electrode arrays permit large-scale electrophysiology by recording from hundreds of neurons simultaneously. We have analyzed the data from Ref. [27, 28] where a neuropixels probe was used to record the activity from the visual cortex, hippocampus and parts of the thalamus in a head-fixed mouse with forepaws resting on a wheel that could move laterally. The dataset contains 242 isolated neurons for which we analyzed the assembly structure using our Bayesian method.

In order to carry out the analysis of the assemblies from large-scale electrophysiological recordings, we need the neuronal activity matrix as input of our inference method. Although it would be natural to define the state of a neuron according to the presence or absence of a spike at any given time point, this definition is not suitable in this context because the synchronous activation of a neuronal population occurs at a much slower time scale than a single spike. Here we define the state of a neuron based on the firing rate instead of single spikes. Periods of increased firing rate due to reverberating activity of recurrent networks can indeed last for tens of seconds, matching the time scale of assembly activity [29].

We extracted the transients of high firing frequency for each neuron by counting the spikes in bins of 0.6s along the recording and then applying the HMM method to detect exponentially-decaying transients (Fig 8A). By applying our inference method to the binary activity matrix representing the states of all 242 neurons over time (Fig 8B), we found that 33% of those neurons can be assigned with 99% confidence to four assemblies satisfying the same constraints on activity, synchrony and asynchrony as applied to calcium imaging data. In particular assemblies 1 and 3 displayed a significantly positive correlation with wheel velocity (reflecting lateral mouse movement) and assembly 2 was weakly anti-correlated with wheel velocity (Fig 8E and 8F). The observation of neuronal assemblies either positively or negatively correlated to the motor output is consistent with the analogous results obtained by analyzing functional imaging data of the mouse cortex.

### Discussion

In this work we introduced a model-based approach to detect neuronal assemblies. Our method is novel in a number of respects. Firstly, it is specifically designed to analyze neuronal population activity by directly making use of features that are known to exists in such datasets (levels of within-assembly synchrony and asynchrony and variation in assembly size and activity patterns) as criteria for grouping neurons into assemblies. Secondly, assignment of neurons to assemblies and characterization of assembly features such as activity rate, and the degree of synchronous and asynchronous firing are estimated simultaneously. Thirdly, because our method is grounded on statistical inference, the level of uncertainty about each of these estimates is quantified. Fourthly, arbitrary definitions of time frames that characterize synchrony are not required because our inference method can extend the time window in which an assembly is active, accounting for sustained periods of activity. This allows us to accommodate assemblies where synchrony occurs over varying periods of time rather than single time points. Finally, we show that within the testing datasets used in this work, our method outperforms a

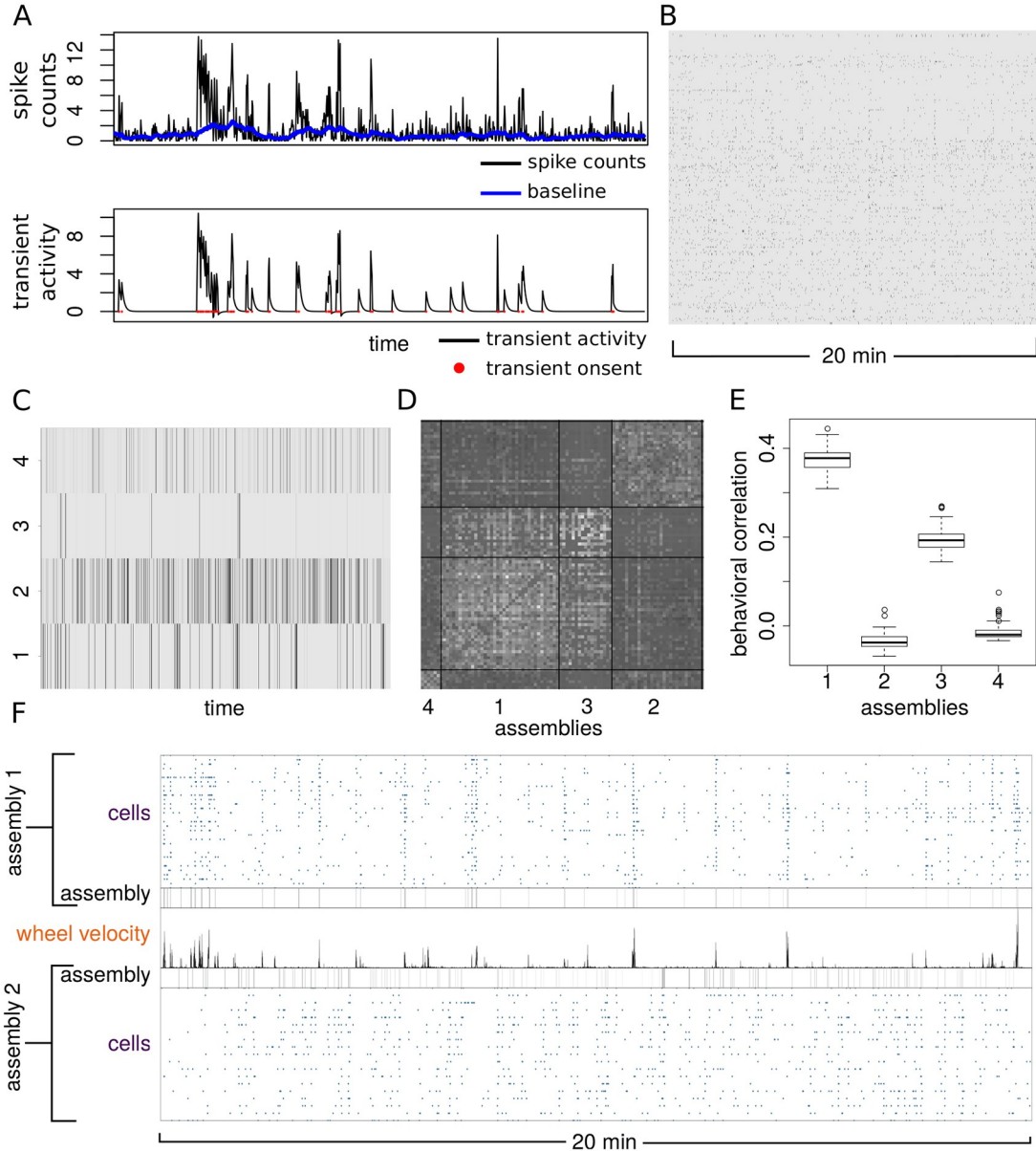

**Fig 8. Assemblies from neuropixels recording of mouse cortex.** (A) Identification of exponentially decaying activity transients from spike counts using bin size of 0.6s. (B) Binary activity matrix obtained by combining transient events from 242 neurons. (C) Raster plot of the assembly activity. (D) Raster plot of the time correlation matrix between neurons sorted by assembly membership. (E) Time correlation between each assembly activity and the absolute wheel velocity. Boxplots represent the distribution of correlation across posterior samples. (F) Comparison between correlated and anti-correlated assemblies with respect to wheel motion.

number of existing techniques over a broad range of sample size, assembly number and assembly features.

Our method defines assemblies of varying size, activity rate and levels of synchronous and asynchronous activity, enabling users to use these features to select assemblies for further analysis. For instance, in our analysis of the zebrafish tectum we focused on assemblies with a high level of synchrony with at least 5 neurons. While there is potential for significant diversity in the assemblies that meet these criteria we show instead that there is a high level of biological

reproducibility when comparing assembly features across fish. An additional finding is that the majority of the assemblies are spatially compact and can be grouped into segregated subnetworks restricted to either the anterior or posterior half of a tectal hemisphere. These findings are consistent with data showing that stimulation of anterior or posterior tectum drives different (approach- and avoidance-like) behaviors. Therefore, the functional segregation of subnetworks of assemblies may be relevant for understanding how activation of different tectal locations leads to mutually exclusive behaviors [30–32]. The assembly activity matrix, obtained from our method, can also be used to identify temporal patterns of assembly activation sequences [33] that might be more relevant than single assembly firing events for understanding tectally-mediated behavior.

Our analysis also reveals that a large fraction of the tectal neurons could not be assigned into a specific neuronal assembly. These neurons may be part of assemblies which were inactive during our recording period. This may be due to chance or there may be some tectal assemblies which are only recruited during sensory stimulation and/or behavior. Alternatively, some of these neurons might not belong exclusively to one assembly. Although our generative model assumes that each neuron belongs exclusively to one assembly, the multimodality in the posterior membership distributions may reflect the presence of such "promiscuous" neurons. A final interpretation is that these neurons are only weakly coupled to population firing. Such neurons have been identified in the mouse and macaque visual cortex where they contrast with neurons whose firing is strongly coupled to the overall firing of the population [34].

We illustrated the general applicability of our inference method by analyzing neuronal assemblies present in functional imaging data and neuropixels recordings from mouse. By exploiting the behavioral information recorded alongside neuronal activity, we have found that some of the assemblies detected with our method were correlated or anti-correlated with specific motor outputs such as running or pupil contraction. These findings agree well with those of the Stringer et al [19, 26], who show using principal component analysis that the first principal component correlates with arousal (as indicated by running speed). Our method builds on these findings by revealing the specific neural assemblies that deferentially correlate with arousal at different time points.

Our method takes binarized neuronal activity as input to estimate the underlying assembly structure. To analyze calcium imaging data and neuropixels recordings we employed a hidden Markov model to extract the onset of activity transients, however the application of our method for detecting assemblies is independent of the binarization step. Users can use different techniques to detect events of increased neuronal activity which depends on the biophysics of the probe or the experimental settings used, and then apply our method to detect assembly structure in the data.

In general, all clustering techniques make specific assumptions on the data features which define classes. Our approach is based on defining those features specifically for neuronal data within a generative model that accommodates minimal hypotheses on the structure of neuronal activity. We believe that any clustering analysis should be grounded on prior hypotheses. The conceptual workflow used in this work involves (1) the development of a generative model where all data assumptions are explicit, (2) the use of a non-parametric method to accommodate variable number of groups and (3) the application of MCMC sampling method to estimate model variables from the posterior distribution. This workflow will likely serve as a model for future generalizations of our technique.

We have shown that our technique can be used as a general tool to study neuronal population data from various species acquired using various methods but we identify some key areas where the method may be particularly valuable. Firstly, because our method accurately assigns neurons to an assembly with statistical confidence it will be particularly useful for

understanding how identified neurons or neuronal types participate in, and contribute to population activity. Furthermore, our method simultaneously quantifies key features of assembly dynamics such as assembly on- and off states, and levels of synchronous and asynchrounous firing, which together reflect the degree of functional coupling between an assembly and its constituent neurons. We anticipate that these measures may be used to quantify how within- and between-assembly interactions are remodeled as a function of development, experience, internal states, and neurological disorder.

## Materials and methods

### Ethics statement

This work was approved by the local Animal Care and Use Committee (King's College London) and was performed in accordance with the Animals (Scientific Procedures) Act, 1986, under license from the United Kingdom Home Office Licence number P9090AEFD. All primary data included in the manuscript came from the use of zebrafish larvae. All procedures were non invasive and classified as mild according to the Animals Act 1986 and as defined by the United Kingdom Home Office, in order to minimize animal suffering. At the end of regulated procedures animals were culled using a schedule 1 method (terminal dose of MS222).

### Zebrafish larvae

For functional imaging experiments we used transgenic zebrafish, Tg(*elavl3:H2B-GCaMP6s*), expressing the nucleus-targeted calcium indicator GCaMP6 under the *elavl3* promoter, which provides near-panneuronal expression [23] (gift from Misha Ahrens, Janelia Research Campus). Larvae were raised at 28.5˚C in Danieau solution and were exposed to a 14 hour ON/10 hour OFF light/dark cycle. Larvae were fed daily from 5 dpf using live rotifers. To maximize optical clarity for imaging Tg(*elavl3:H2B-GCaMP6s*) larvae were crossed with compound *roy; nacre* double homozygous mutants (*casper*) larvae which lack melanocyte and iridophore pigmentation. This work was approved by the local Animal Care and Use Committee (King's College London), and was carried out in accordance with the Animals (Experimental Procedures) Act, 1986, under license from the United Kingdom Home Office.

### Volumetric calcium imaging

At 7 days post fertilization (dpf) zebrafish were mounted in 2% low melting point agarose in Danieau water with their dorsal side facing up on a custom built imaging slide and were submerged in Danieau water. These fish were left for 1 hour in the light so that the fish could settle, reducing drift while imaging. Prior to imaging the fish were placed under the microscope objective in total darkness for 30 minutes to allow the fish to adjust to the imaging conditions. On-going activity was monitored by imaging the calcium dynamics of thousands of neurons in both tectal hemispheres for 1 hour with a custom built 2-photon microscope (Independent NeuroScience Services, INSS). Excitation was provided by a Mai Tai HP ultrafast Ti:Sapphire laser (Spectraphysics) tuned to 940nm. Laser power at the objective was kept below 15 mW for all fish. Emitted light was collected by a 16x, 1 NA water immersion objective (Nikon) and detected using a gallium arsenide phosphide (GaAsP) detector (ThorLabs). Images (256 x 256 pixels) were acquired at a frame rate of 60Hz by scanning the laser in the x-axis with a resonant scanner and in the y-axis by a galvo-mirror. To enhance signal-to-noise every 2 frames for each focal plane were averaged. The focal plane was adjusted in 15$\mu$m steps using a piezo lens

holder (Physik Instrumente). This allowed for volumetric data consisting of 5 focal planes to be collected at a volume rate of 4.8Hz. Scanning and image acquisition were controlled by Scanimage Software (Vidrio Technologies).

## Image registration

Any volumetric stacks where large numbers of neurons in the imaging plane drifted out of view were discarded. All other stacks were corrected for x-y drift by aligning each every frame in each slice in the stack independently. First every frame for each slice was aligned to its first frame, then they were aligned again to the mean of these images. This registration was performed using a non-rigid body alignment algorithm contained within the SPM8 package for MATLAB (http://www.fil.ion.ucl.ac.uk/spm/software/spm8).

## Segmentation of tectal cells

After image registration, images were processed using custom-made C++ software of cell segmentation based on a flooding algorithm. For each plane in the volume, we generate the time average of the image and apply a Gaussian filter to smooth the spatial details below the size of a cell. Next, we applied the following segmentation algorithm to extract single cell time sequences. Starting from the top fluorescence $f^{(max)}$ we sequentially label voxels according to the nearest segmented cell within a radius compatible with the size of a cell. To take time correlations into account we introduced a distance between voxels $h(V, V')$ which combines physical separation and time correlation as

$$h(V, V') \equiv \sqrt{(V_x - V'_x)^2 + (V_y - V'_y)^2 + \eta C_{VV'}} \tag{28}$$

where the parameter $\eta$ introduce some flexibility in the importance of the Pearson's time correlation $C_{VV'}$ between the time sequences of $V$ and $V'$. Whenever a new label is assigned, the corresponding voxel becomes the representative of the new cell and the distance between a voxel $V$ and a cell is obtained by Eq (28) by replacing $V'$ with the voxel which representing the cell.

**Algorithm 3** Segmentation algorithm
```
1: Set f = f^(max)
2: while f ≥ f^(min) do
3:   for each voxel V with fluorescence ≥ f do
4:     Calculate the distance h between V and its nearest cell accord-
ing to Eq (28)
5:     if h < h^(max) then
6:       Assign V to the nearest cell
7:     else
8:       Assign V to a new label
9:   f = f - Δf
```

## Detection of calcium transients

Let us consider the fluorescence trace $\{X_k\}_{k=0}^{M}$ where $k$ denotes the time index running from 0 to $M$. As discussed in the main text, we decompose fluorescence traces into the sum of calcium transient $c_k$, baseline activity $b_k$ and a source of Gaussian noise. The hidden Markov model is

summarized by the set of equations

$$s_k \quad \sim \quad \text{Bernoulli}(q\delta t) \tag{29}$$

$$b_k|b_{k-1} \quad \sim \quad \mathcal{N}(b_{k-1}, \sigma_B\sqrt{\delta t}) \tag{30}$$

$$c_k|c_{k-1}, s_k \quad \sim \quad \begin{cases} \mathcal{N}(c_{k-1}e^{-\lambda\delta t}, \sigma_C) & s_k = 1 \\ \delta(c - c_{k-1}e^{-\lambda\delta t}) & s_k = 0 \end{cases} \tag{31}$$

$$x_k \quad \sim \quad \mathcal{N}(b_k + c_k, \sigma_x) \tag{32}$$

where calcium transients decay exponentially with decay constant $\lambda$ while their onset is specified by the latent variable $s_k$ representing the hidden on/off state of a neuron, modeled as a Bernoulli process with rate $q$. $\delta t$ is the time interval between samples. To obtain an estimate of the hidden variables at all times we maximize iteratively the probability of the latent state at time $k$, given the state at $k-1$ and the observed value $x_k$

$$
\begin{aligned}
\{\hat{s}_k, \hat{c}_k, \hat{b}_k\} \quad &= \quad \underset{\{s_k, c_k, b_k\}}{\text{argmax}} \, P(s_k, c_k, b_k | \hat{s}_{k-1}, \hat{c}_{k-1}, \hat{b}_{k-1}, x_k) = \\
&= \quad \underset{\{s_k, c_k, b_k\}}{\text{argmax}} \, P(s_k, c_k, b_k | \hat{s}_{k-1}, \hat{c}_{k-1}, \hat{b}_{k-1}) \cdot P(x_k | c_k, b_k) \\
&= \quad \underset{\{s_k, c_k, b_k\}}{\text{argmax}} \, P(s_k) P(c_k | s_k, \hat{c}_{k-1}) P(b_k | \hat{b}_{k-1}) \cdot P(x_k | c_k, b_k).
\end{aligned}
\tag{33}
$$

where all the probabilities can be obtained from the definition of the model

$$P(x_k|c_k, b_k) = \frac{1}{\sqrt{2\pi\sigma_x^2\delta t}} \exp\left\{ -\frac{(x_k - c_k - b_k)^2}{2\sigma_x^2} \right\} \tag{34}$$

$$P(b_k|b_{k-1}) = \frac{1}{\sqrt{2\pi\sigma_B^2\delta t}} \exp\left\{ -\frac{(b_k - b_{k-1})^2}{2\sigma_B^2\delta t} \right\} \tag{35}$$

$$P(c_k|s_k, c_{k-1}) = \begin{cases} \frac{1}{\sqrt{2\pi\sigma_C^2\delta t}} \exp\left\{ -\frac{(c_k - c_{k-1}e^{-\lambda\delta t})^2}{2\sigma_C^2\delta t} \right\} & \text{if } s_k = 1 \\ \delta(c_k - c_{k-1}e^{-\lambda\delta t}) & \text{if } s_k = 0 \end{cases} \tag{36}$$

$$P(s_k = 1) = q\delta t \tag{37}$$

We can obtain analytical expression for the estimated calcium and baseline at each time by considering separately the two cases $s_k = 0$ or $s_k = 1$.

- **case** $s_k = 0$. To maximize the probability we need to maximize the function

$$g_0(b_k) = \frac{1}{\sqrt{2\pi\sigma_x^2}} \frac{1}{\sqrt{2\pi\sigma_B^2\delta t}} \exp\left\{ -\frac{(x_k - c_k^* - b_k)^2}{2\sigma_x^2} - \frac{(b_k - b_{k-1})^2}{2\sigma_B^2\delta t} \right\} \tag{38}$$

where we defined $c_k^* = c_{k-1}e^{-\lambda\delta t}$. By setting $\frac{d\log g_0}{db_k} = 0$ we get

$$\frac{1}{\sigma_x^2}(x_k - c_k^* - b_k) - \frac{1}{\sigma_B^2\delta t}(b_k - b_{k-1}) = 0 \tag{39}$$

therefore the values of $c_k$ and $b_k$ which maximize the log-likelihood when $s_k = 0$ are

$$\hat{c}_k^{(0)} = c_k^* \tag{40}$$

$$\hat{b}_k^{(0)} = \frac{\frac{1}{\sigma_x^2}\left(x_k - c_k^*\right) + \frac{b_{k-1}}{\sigma_B^2 \delta t}}{\frac{1}{\sigma_x^2} + \frac{1}{\sigma_B^2 \delta t}} \tag{41}$$

- **case** $s_k = 1$. In the case of a calcium transient we need to maximize the function

$$g_1(c_k, b_k) = \frac{1}{\sqrt{2\pi\sigma_x^2}} \frac{1}{\sqrt{2\pi\sigma_B^2 \delta t}} \frac{1}{\sqrt{2\pi\sigma_C^2}} e^{-\frac{(x_k - c_k - b_k)^2}{2\sigma_x^2} - \frac{(b_k - b_{k-1})^2}{2\sigma_B^2 \delta t} - \frac{(c_k - c_k^*)^2}{2\sigma_C^2}} \tag{42}$$

analogously to the case $s_k = 0$, by maximizing $\log g_1$ with respect to both $c_k$ and $b_k$ to zero we get

$$\hat{c}_k^{(1)} = c_k^* + \frac{x_k - c_k^* - b_{k-1}}{1 + \frac{\sigma_B^2 \delta t}{\sigma_C^2} + \frac{\sigma_x^2}{\sigma_C^2}} \tag{43}$$

$$\hat{b}_k^{(1)} = b_{k-1} + \frac{\sigma_B^2 \delta t}{\sigma_C^2}\left(\hat{c}_k^{(1)} - c_k^*\right) \tag{44}$$

By combining the two conditions we have

$$\{\hat{s}_k, \hat{c}_k, \hat{b}_k\} = \begin{cases} \{0, \hat{c}_k^{(0)}, \hat{b}_k^{(0)}\} & \text{if } (1 - q\delta t) \cdot g_0(\hat{b}_k^{(0)}) > q\delta t \cdot g_1(\hat{c}_k^{(1)}, \hat{b}_k^{(1)}) \\ \{1, \hat{c}_k^{(1)}, \hat{b}_k^{(1)}\} & \text{if } (1 - q\delta t) \cdot g_0(\hat{b}_k^{(0)}) < q\delta t \cdot g_1(\hat{c}_k^{(1)}, \hat{b}_k^{(1)}) \end{cases} \tag{45}$$

To complete the iterative algorithm we have to assign values to the initial calcium and baseline level. A simple choice for the initial state is $b_0 = x_0$ and $c_0, s_0 = 0$ however other criteria may be more effective if transients are expected to occur at the beginning of the recording.

This procedure can be applied to any time sequence to estimate the onset of exponentially decaying transient, however it requires the variance parameters $\sigma_x$, $\sigma_B$ and $\sigma_C$ as well as the decay constant $\lambda$ and the Bernoulli rate $q$. Furthermore, using the same set of parameters for all traces might lead to estimation biases. To account for parameter variation across cells we employed a "plug-in" refinement method similar to the $k$-means algorithm. Starting from an initial guess of the parameters we apply the two steps

1. uenerate estimates of $\{s_k, c_k, b_k\}$ using Eq (45)

2. Use $\{s_k, c_k, b_k\}$ to obtain new parameters according to

$$\sigma_B^2 = \frac{1}{M} \sum_{k=1}^{M} \frac{(b_k - b_{k-1})^2}{\delta t} \tag{46}$$

$$\sigma_C^2 = \frac{1}{M_1} \sum_{k:s_k=1} (c_k - c_k^*)^2, \quad M_1 = \sum_k s_k \tag{47}$$

$$\sigma_x^2 = \frac{1}{M} \sum_k (x_k - c_k - b_k)^2 \tag{48}$$

$$q = \frac{M_1}{M \delta t} \tag{49}$$

Because calcium and baseline estimated according to Eq (45) are an approximation of the true maximum-likelihood trajectory, repeating these two steps might lead to singular values of the parameters. The first iterations of this plug-in method however provide a better decomposition than using fixed parameters. The binary activity matrices used as input for our inference method were generated by iterating twice the steps 1 and 2.

## Conjugate beta priors

To calculate the marginal likelihood in Eq (14) we exploit the conjugate character of the beta priors on the continuous parameters with respect to the model likelihood. This feature allows us to easily solve integrals of the form

$$
\begin{aligned}
\int_0^1 p^{N_1}(1-p)^{N_2} \text{Beta}(p; \alpha, \beta) &= \int_0^1 \frac{1}{B(\alpha, \beta)} p^{N_1 + \alpha - 1}(1-p)^{N_2 + \beta - 1} = \\
&= \frac{B(N_1 + \alpha, N_2 + \beta)}{B(\alpha, \beta)}
\end{aligned}
\tag{50}
$$

where $B(\cdot, \cdot)$ is the Euler beta function.

## Dirichlet process through Metropolis-Hastings rule

In the main text we employed the Metropolis-Hastings rule introduced by Neal in Ref. [21] to introduce the Dirichlet process prior as part of a Monte Carlo sampler. The general model discussed by Neal is

$$y_i | \theta_i \quad \sim \quad F_{\theta_i} \tag{51}$$

$$\theta_i | G \quad \sim \quad G \tag{52}$$

$$G \quad \sim \quad DP(G_0, \alpha) \tag{53}$$

where $\theta_1, \cdots, \theta_n$ are the parameters specifying the conditional distribution $F$ of each data point $y_1, \cdots, y_n$. $G$ is a measure drawn from the Dirichlet process (DP) with base measure $G_0$ and concentration $\alpha$. With this setting, when reassigning the value $y_i$ from the old cluster $\mu_0$ to the proposed cluster $\mu^*$, the Metropolis-Hastings rule reads

$$a(\mu^*, \mu_0) = \min\left(1, \frac{F_{\mu^*}(y_i)}{F_{\mu_0}(y_i)}\right). \tag{54}$$

For our model of neuronal assemblies, each data point $y_i$ corresponds to the entire time sequence of neuron $i$, $s_{i1}, \cdots, s_{iM}$ while the parameters correspond to the time sequences of the corresponding assembly $\omega_{ti}1, \cdots, \omega_{ti} M$. The ratio in Eq (54) corresponds to the likelihood ratio between assigning the data point $y_i$ to the new group or the old one. In the collapsed version of our model, i.e. after integrating out the continuous parameters, the data (neuronal states $s_i$ over time) are no longer conditionally independent given their cluster parameters

(assembly states $\omega$) as they are in Eq (51). However, we can still calculate the analog of the likelihood ratio of Eq (54) by considering the second factor in Eq (16) (excluding the multinomial component which is replaced by the Dirichlet process prior)

$$P(s|t, \omega) = \prod_{\mu=1}^{A} \prod_{z \in \{0,1\}} \frac{B(T_\mu^{z1}, T_\mu^{z0})}{B(\alpha_z^{(\lambda)}, \beta_z^{(\lambda)})}, \tag{55}$$

from which we can obtain the ratio in Eq (26).

## Generalization of pairwise assignment matrices to multiple membership

The pairwise assignment matrices $I$ used to quantify the performance of different methods are defined as

$$I_{ij}^{(\tau)} = \begin{cases} 1 & \text{if } \tau(i) = \tau(j) \\ -1 & \text{if } \tau(i) \neq \tau(j) \end{cases}. \tag{56}$$

For neurons with multiple memberships the assignments $\tau(i)$ are sets of assemblies. We generalized the pairwise assignment matrices in Eq (56) by setting to 1 elements corresponding to pair of neurons sharing at least one assembly membership, and -1 otherwise, namely

$$I_{ij}^{(\tau)} = \begin{cases} 1 & \text{if } \tau(i) \cap \tau(j) \neq \emptyset \\ -1 & \text{if } \tau(i) \cap \tau(j) = \emptyset \end{cases}. \tag{57}$$

## Standard clustering algorithms

In the main text we discuss the comparison between our Bayesian inference method and three clustering algorithms based on commonly used techniques such as: $k$-means, PCA and spectral clustering. We have used the $k$-means clustering algorithm implemented in R. For each condition of asynchrony, number of cells and assemblies we used 20 different initializations of $k$-means to avoid local maxima and kept the classification that maximized the between cluster variance. To estimate the number of clusters we used the commonly used silhouette method.

For the PCA-based clustering method we first obtained a "shuffled model" of the data by applying a random time shift to each neuron (circular shuffling) drawn from a uniform distribution between 0 and 200 and then extracting the eigenvalues of the covariance matrix across shuffled neurons. By repeating this procedure 200 times we compared the eigenvalues obtained from PCA analysis of the original neuronal activity matrix with the average eigenvalues of shuffled data. We consider principal components significant when the corresponding eigenvalue was 2 standard deviation above the shuffled model. The number of significant components is used as an estimate of the number of neuronal assemblies. We then applied the varimax rotation on these significant principal components and assigned neuronal memberships by labeling each neuron with the index of maximum absolute loading in the varimax-rotated components. Only neurons with maximum loading one standard deviation above the mean loading were assigned to an assembly, while the remaining were considered as independent (free neurons).

We adapted the pipeline introduced in Ref. [5] to our testing datasets. This pipeline employs spectral clustering with graph-theoretic estimation of the number of assemblies. In the main text we refer to this method as spectral clustering but the analysis involves several steps. (1) Select synchronous activity frames by requiring the overall activity to be 2 standard

deviation above a shuffled model obtained by permuting over time each neuronal activity; (2) build a k nearest neighbor graph of synchronous frames from their cosine similarity; (3) use the Newman-Reinart method to estimate the number of communities $K$ in the graph based on the degree-corrected stochastic block model [22]; (4) perform spectral clustering with $K$ clusters; (5) count the times at which each neuron participates to any of the synchronous frames in each cluster and extract neuronal identities as the cluster index with larger counts. Neurons with affinity lower than 20% were considered as not belonging to an assembly (free neurons).

## Graph theoretic analysis of Pearson's correlations

To carry out the time correlation analysis between assemblies in the fish tectum, for each posterior sample of the matrix $\omega$ representing the activity of each assembly by row, we calculated the Pearson's correlation between all rows. The graph describing the assembly interaction was constructed by adding an edge between all pair of assemblies which had a positive correlation in over 95% of the posterior samples. The Girvan-Newman algorithm [35] implemented in the igraph [36] R package (`edge.betweenness.community` function) was used to identify the components of the correlation graph.

## Runtime of the Gibbs sampler

The CPU time required to analyze a data set of approximately 1000 neurons for 1000 time frames is 2-3 hours on a regular desktop machine. All our analyses were done using an iMac with CPU intel(R) core(TM) i7-7770 3.40GHz. Larger data sets with twice as many neurons and time frames can take up to 12h.

## Supporting information

**S1 Fig. Graphical representation of the generative model.** (A) Full model. Nodes in the network represent model variables, and edges denote statistical dependency, (B) collapsed model where assembly activation probability ($p$) and conditional probability of neuronal activation ($\lambda$) are integrated out, (C) collapsed model with Dirichlet process prior on the number of assemblies.
(TIFF)

**S2 Fig. Increasing assembly activity reduces the volume of the non-detectable phase.** We repeated the analysis represented in Fig 2 to determine the effect of assembly activity on the phase diagram of assembly detectability. When assembly activity increases, the non-detectable regime shrinks towards the line $\lambda(0) = \lambda(1)$, corresponding to the limit where the data are non longer informative about neuronal identity.
(TIFF)

**S3 Fig. Distribution of activity, synchrony and asynchrony levels.** (A) Scatter plot representing the distribution of synchrony and asynchrony levels within one fish. Green (dashed) lines display the thresholds applied to activity (0.005, corresponding to $\approx$1 event per minute) and coherence (0.05). (B, C) Distribution of synchrony and asynchrony levels versus activity for all assemblies from all fish.
(TIFF)

**S4 Fig. Assembly vs cell activity.** Comparison between the binary activity of all tectal neurons belonging to assembly 48 in Fig 6 (top), the activity of the assembly (middle) and the fraction

of active cells within the specific assembly over time (bottom).
(TIFF)

**S5 Fig. Differences between highly correlated assemblies.** Comparison between assemblies of cortical neurons 15 and 32 from Fig 7. Although both assemblies correlate with running speed, member neurons are separated in two groups due to their different levels of activity, synchrony and asynchrony.
(TIFF)

**S6 Fig. Correlation analysis of assembly and free neurons.** Comparison between the statistics of the time correlation between each assembly neuron and the assembly it belong to (orange), and the maximal correlation between free neurons and the assemblies determined with our method (blue). The correlations of assembly neurons is significantly shifted to higher values with respect to free neuron maximal correlations.
(TIFF)

**S7 Fig. Assembly compactness.** Comparison between the distribution of assembly spatial extension across each fish data (orange) against a null distribution obtained from random groups of neurons of equal size (green). This analysis confirms that tectal assemblies are significantly more compact (reduced extension) than random groups of neurons. The assembly spatial extension was calculated as the area $E$ of the ellipse fitting the assembly 2D distribution, $E = \pi \sqrt{\xi_1 \xi_2}$, where $\xi_1$ and $\xi_2$ are the two eigenvalues of the covariance matrix of the XY neuronal coordinates within the assembly.
(TIFF)

**S8 Fig. Comparison of Bayesian inference performance to $k$-means, PCA and spectral clustering using testing datasets with single membership neurons.** (A) Performance comparison across levels of asynchrony. Dots correspond to independently generated data sets while solid lines show the average performance for each method over all simulated data. (B) Comparison across number of assemblies. (C) Comparison across number of neurons. (D-F) Standard deviation of the performance across simulated data per parametric condition. Unless specified otherwise, surrogate datasets were generated using 400 neurons and 1000 time frames distributed over 5 assemblies with assembly activity of 5%, synchrony 50% and asynchrony 10%. For $k$-means we used a number of clusters estimated according to the silhouette method. The number of significant principal components in PCA clustering was obtained by the circular shuffling method whereas for the spectral clustering we used the Newman-Reinart graph-theoretic community detection method [22] (see Materials and methods). The performance was measured according to Eq (27) by comparing each set of assignments with ground truth assignments.
(TIFF)

**S1 Video. Zebrafish tectum.** Recording of calcium activity from the zebrafish tectum throughout 5 planes at 15$\mu$m apart with an acquisition frequency of 4.8Hz per volume.
(AVI)

**S2 Video. Assembly distributions.** Visualization of 3D distributions of tectal assemblies obtained from one fish.
(AVI)

**S3 Video. Assembly dynamics.** Assembly dynamics within a single plane with color coded neurons according to assembly membership.
(AVI)

**S4 Video. 3D distributions of cortical assemblies.** Most common 3D distribution distributed across the cortical volume.
(AVI)

**S5 Video. Column distributions of cortical assemblies.** Representative cortical assembly spanning through the *z*-axis.
(AVI)

**S6 Video. Ventral assemblies in the cortex.** Representative cortical assembly limited to ventral planes.
(AVI)

## Acknowledgments

This work is dedicated to the memory of Diego Diana. The authors would like to acknowledge QueeLim Ch'ng and Setsuko Sahara (King's College London) for providing HPC machines used for data analysis. Diana Passaro (Francis Crick Institute, London), Marco Banterle (London School of Hygiene and Tropical Medicine), Juan Burrone, Adil Khan and Marco Bazo (King's College London), Marcus Triplett, Jan Mölter and Geoffrey Goodhill (Queensland Brain Institute, Brisbane) for discussions and comments on the manuscript. Misha Ahrens (HHMI Janelia) for providing Tg(*elavl3:H2B-GCaMP6s*) zebrafish. Carsen Stringer (HHMI Janelia) and Nicholas Steinmetz (University College London) for providing functional imaging data and neuropixels recordings of the mouse cortex.

## Author Contributions

**Conceptualization:** Giovanni Diana, Martin P. Meyer.

**Data curation:** Thomas T. J. Sainsbury.

**Formal analysis:** Giovanni Diana.

**Funding acquisition:** Martin P. Meyer.

**Investigation:** Thomas T. J. Sainsbury.

**Methodology:** Giovanni Diana.

**Project administration:** Martin P. Meyer.

**Resources:** Martin P. Meyer.

**Software:** Giovanni Diana.

**Supervision:** Giovanni Diana, Martin P. Meyer.

**Validation:** Thomas T. J. Sainsbury.

**Visualization:** Giovanni Diana.

**Writing – original draft:** Giovanni Diana, Thomas T. J. Sainsbury, Martin P. Meyer.

**Writing – review & editing:** Giovanni Diana, Martin P. Meyer.

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
