## [Decision Letter · Decision Letter 0]

20 Jul 2019

Dear Dr Diana,

Thank you very much for submitting your manuscript, 'Bayesian inference of neuronal ensembles', to PLOS Computational Biology. As with all papers submitted to the journal, yours was fully evaluated by the PLOS Computational Biology editorial team, and in this case, by independent peer reviewers. The reviewers appreciated the attention to an important topic but identified some aspects of the manuscript that should be improved.

We would therefore like to ask you to modify the manuscript according to the review recommendations before we can consider your manuscript for acceptance. Your revisions should address the specific points made by each reviewer and we encourage you to respond to particular issues Please note while forming your response, if your article is accepted, you may have the opportunity to make the peer review history publicly available. The record will include editor decision letters (with reviews) and your responses to reviewer comments. If eligible, we will contact you to opt in or out.raised.

- Supporting Information uploaded as separate files, titled 'Dataset', 'Figure', 'Table', 'Text', 'Protocol', 'Audio', or 'Video'.

We hope to receive your revised manuscript within the next 30 days. If you anticipate any delay in its return, we ask that you let us know the expected resubmission date by email at ploscompbiol@plos.org.

Sincerely,

Claus C Hilgetag

Methods Editor

PLOS Computational Biology

Claus Hilgetag

Methods Editor

PLOS Computational Biology

[LINK]

Reviewer's Responses to Questions

**Comments to the Authors:**

Reviewer #1: In the manuscript, Diana et al. propose an alternative method for the analysis of neuronal populations based on a Bayesian approach. This technique enables a statistical analysis of neuronal ensemble properties without using dimensionality reduction approaches. The authors used artificial data to validate the methodology and compare with other approaches (k-means, PCA / factor analysis, spectral clustering), and real data obtained from the zebrafish optic tectum and mouse cortex using two-photon calcium imaging and from various regions of the mouse brain using neuropixel electrodes.

This is indeed an additional alternative method among those already developed to study population dynamics in the brain.

I am convinced that this approach will be useful for neurobiology labs studying population dynamics. This approach is suitable for a variety of recording techniques and animal models. Although I believe the manuscript is suitable for publication in Plos CB, I have some concerns that need to be addressed:

Comments:

1) In my opinion, comparison of one method with others is always problematic since a method could perform better for a given aspect and be worse for a different one. Also, the authors use different arbitrary parameters and thresholds for each of the different methods. Since the performance of these methods depends on the chosen parameters, the comparison between methods is not very reliable. This counts also for reference [16] which is cited by the authors. So, I would be cautious on how to interpret these comparisons. I believe that there is no need for justification. The decision needs to remain on the potential users who will decide which method will be best for analysing their own data. The authors do not mention CPU time of their approach. How long does it take to analyse their data ? It can be done with a regular computer or a computer cluster is necessary ?

2) line 102. The claim that each neuron belongs to only one ensemble is too strong and not realistic. The authors are aware of this as they explicitly mention it in line 466. I would definitely incorporate their suggestion of introducing latent variables. This could indeed explain why 50% of the neurons in the tecum of the fish behave as isolated neurons. Do these neurons show pairwise correlations with others ? This should be tested to check whether these neurons are indeed isolated form the rest or are excluded because they belong to more than one ensemble. I would also use synthetic datasets where ensembles will not be discrete (neurons should belong to more than one ensemble).

3) All figure legends are very brief, especially those of the supplementary figures. Legends should describe the figures in detail for non-expert readers. Some axes legends are missing. Legend of Figure S6 is missing.

4) Adding an explanatory figure on how the HHM approach for detecting the calcium transients will help better understanding the method.

5) Fig 5E. is this a stack or a single plane? please specify. Fig. 6F is confusing. Adding just a few it will be enough. The rest can be shown in a supplementary figure.

6) Why only time frames with more than 15 active cells were considered? please justify or explain.

7) explain the rationale of using the parameters in lines 323-328.

8) line 450. The authors claim that the ensembles are compact but they do not quantify compactness.

9) Since this is a methods manuscript, it is important that the reviewers could test the approach. The codes are no available. Also, it is necessary to clearly and in details describe all the equations and the mathematical procedures. As it is is sometimes difficult to follow. This can be done in the supplementary information. For example, in line 203 why Gmu ^( -i) ?

10) Statistics in Fig 3 are missing.

11) Figure 7A is very confusing. I suggest coloring only neurons that belong to the ensembles shown in Fig 7D and E and Fig S5 and S6.

12) line 336. the temporal resolution was changed from the original recording as only time frames with more than 15 active cells were used for analysis. What's the number of frames excluded?

13) Ensemble usually refers to a group of neurons that could be co-activated by a common source or stimulus. Assembly refers to a group of neurons that are activated together because they are interconnected. Since the authors used spontaneous activity to identify the groups of neurons I would change ensemble to assembly.

14) theory. page 3. the notation is confusing.

Reviewer #2: In their manuscript, Diana et al. present a new approach to identify ensemble activity in neuronal population data. Unlike previous methods this approach is purely model-driven and relies on statistical inference using a generative model of ensemble activity. The authors validate their approach on test data generated from their generative model and test its performance against other methods, claiming that their approach is superior to the others. Finally, they demonstrate the applicability of their approach to calcium imaging data from the optic tectum of larval zebrafish and the visual cortex of mice and show that ensemble activity tends to be spatially localised and that the activity of different ensembles might have behavioural correlates.

Overall, I think the manuscript presents an interesting and promising new method to identify neuronal ensembles in neuronal population activity. I am only concerned about the comparison between the authors' approach and other clustering algorithms that have been applied in a way that seems not always appropriate and fair. Apart from that my remaining comments in the following are mostly about a consistent presentation of the results in terms of terminology and mathematical notation.

## Major concerns/comments:

* Line 256ff. / Figure 3: The results seems to suggest that the proposed approach performs perfect under all conditions. However, to some extend this does not seem too surprising given that the data for the assessment was generated from the same generative model that is then used to fit the data. While for a validation this is fine, it seems not fair for a comparison between different methods. The authors would need to show such performance on test data from an independent generative model of ensemble activity.

Also, how where the neurons in the test data distributed into the different ensembles? It seems that the population was simply divided into five (or more) roughly equally sized ensembles. If so, this also raises the question how the performance is affected by having some neurons assigned to individual ensembles and in that sense having neurons that do not contribute to the ensemble activity, i.e. "free neurons" (Figure 5).

* Line 584ff.: When k-means clustering was used, was it performed on all activity frames or only on those that had an ensemble event? In case of the former, the frames that do not have an ensemble event could potentially impact the clustering because all the frames have to be assigned to one of the five (or more) clusters. However, in order to produce an optimal clustering of the ensembles there might be more than the assigned number of clusters necessary in order to be able to cluster the frames that contain no information about ensembles away from those that represent ensemble events.

Furthermore, the k-means algorithm performs the clustering according to some notion of distance. Which distance was used here?

And, given the number clusters that the k-means clustering returned, how was the neuronal identify of the ensembles reconstructed?

* Line 590ff.: When applying the PCA method to infer ensembles, a varimax rotation was used as opposed to the promax rotation (cf. Romano et al. 2015). Why did the authors choose to do so?

Also, when assigning the ensemble membership there might be neurons that have a very low loading in every of the varimax-rotated components. Rather than being forced into an ensemble defined by a component, they should be considered independent.

* Line 601ff.: When applying the spectral clustering method to infer ensembles, it seems that the last step in the procedure is missing in which given the clusters from the spectral clustering activity core ensembles were constructed and frames were reassigned to different clusters (cf. Avitan et al. 2017). Why did the authors choose to do so?

Similarly as in PCA clustering, there might be neurons that have a very low affinity to some of the clusters. However, the procedure chosen here forces them into an ensemble although they should be really considered independent.

## Minor concerns/comments:

* Line 39f.: It is stated that in PCA-base methods the number of ensembles "is equal to the number of principal components required to explain the variance of the data". However I believe that in these methods the question is not really about how much variance is explained by including a certain number of principal components. Rather in these methods ensembles correspond to principal component that cannot be explained by chance, i.e. if there were no temporal correlations between the neurons beyond those that are present given the overall activity level alone (cf. Lopes-dos-Santos et al. 2013).

* Line 52: "a-priori" should be spelled as in the remaining part of the manuscript, "a priori" (Line 94).

* Line 101f.: "We assume that each neuron belongs to one ensemble [...]" - Clearly, in every population there are neurons that do not participate in the ensemble activity in the sense that they largely act independently. I believe that the authors suggest that such neurons are part of their individual ensembles. However, since one could argue that an ensemble requires at least two members, it might be good to mention explicitly that ensembles in their model can also consist of single neurons.

* Line 112ff.: The definition of the ensemble synchrony/asynchrony seems to be uniform across the neurons of this ensemble. However, some neurons might be more reliably activity given the ensemble is active than others and in that sense have a higher affinity to the ensemble (cf. Thompson, Scott 2016). Therefore, how would this assumption affect the applicability of the approach?

* Line 122f.: The terms "coherence" and "noise" are not defined and seem to correspond to the previously introduced terminology of "synchrony" and "asynchrony". This should be fixed.

* Equations (4), (5): It seems that the shape parameters do not depend on the ensemble. However, later when doing the inference (Equations (21)-(23)), there is a differentiation between shape parameters for the different ensembles. Hence, if the authors assume different shape parameters for different ensembles, they should indicate that also in Equations (4), (5) and add the corresponding subscripts.

* Equations (11)-(13): To make the relation of these definitions to the definitions above (Equation (9)) clear, the authors should also state the dependence on z and z' explicitly.

* Line 152f.: Similar to the way it is mentioned before, it should be stated that T^{zz'}_{S;\\mu} also only counts certain events "up to an additive constant".

* Algorithm 1, 2: In order to the denote the ranges the different parameters are chosen from, the same notation as in the rest of the manuscript should be used, i.e. "\\mu \\in \\lbrace 1, \\cdots A \\rbrace" etc.

* Line 218f.: It is stated that ensemble labels are initialised uniformly to values "between 0 to A^{max} where A^{max} \\approx N". First, I believe the range is given by 1 to A^{max}, because by definition 0 is not an admissible label (Line 103). Second, it is not clear what range from which the labels are drawn actually is. I believe that it should maybe say "A^{max} = N".

* Figure 1: "Cells assigned to A \\approx N ensembles are [...]" - It is very unlikely that the initialisation procedure (i.e. drawing labels independently and uniformly for the range of 1 to N) as described before results in approximately N different ensembles. In fact, in expectation there will be N * ( 1 - ( 1 - 1/N )^N ) ensembles which for N sufficiently large means that A \\approx 0.63 N.

* Figure 2 and Figure S2: The relation between synchrony and asynchrony for the recovery of the ensembles is discussed in the main text and the caption. However, in the figure the axis labels read "Ensemble coherence" and "Noise". This should be fixed. Also, I might be helpful to include the corresponding symbols \\lambda(1) and \\lambda(0) in the labels in parentheses.

* Figure 3: The variability in the performance is shown in panels D-F and not only D-E. In addition, while the axis labels read "Performance variance", in the caption this is referred to as the "standard deviation". What is actually shown?

There are characters missing in the axis labels in panel E and F.

* Figure 4: The caption for panel A distinguishes between a left and right part. However, the system setup is not shown in panel A.

* Figure 5: Unlike in the caption, the axis labels in panel D read "coherence" and "noise" which should be "synchrony" and "asynchrony" as before.

* Figure 6: In the network in panel C some edges are drawn in red. What is their relevance?

When measuring the physical distance between neuronal ensembles, how is this done?

* Figure 7: What is the temporal scale in panels B and D?

* Line 416f.: The HMM method to detect the onset of calcium transients was also used to detect the onset of increases in the firing rate. However, the HMM method explicitly assumes an exponential decay of the calcium concentration (Equation (31)). Since the firing rates probably do not have this characteristic, how is this method still applicable?

* Line 560ff.: In the interest of clarity in the presentation, I would suggest that the notation matches the one in the main text. In particular, this means that time is indexed by k running from 1 to M.

* Equation (29): The parameter q seems to be undefined. How is it chosen? Also, from Equation (37) it seems that it should actually be s_{t} \\sim \\operatorname{Bernoulli}(q \\delta t).

* Equation (31): The parameters \\lambda and \\delta t seem to be undefined. How are the chosen? It is only later (Line 571) mentioned that \\lambda is the calcium decay constant.

* Equation (32): I believe it should be a lower case x_{t} given this is how the variable is used mostly throughout this section. However, this should also be fixed in the other instances where the variable is used upper case to be consistent.

Also, instead of = it should be \\sim.

* Equation (36): Given the definition in Equation (31) and e.g. Equation (42), \\delta t should not appear in the variance.

* Line following Equation (38): The derivative should be taken from g_{0}.

* Figure S3: The terms "coherence" and "noise" need to be replaced with "synchrony" and "asynchrony"

* References: The author might want to consider revising the references as some author names are misspelled and some DOIs do not show fully qualified DOIs but rather DOI URLs.

Reviewer #3: In this work the authors describe a new algorithm to identify the neural ensembles in neurophysiological data. They achieve this by inverting hierarchical generative model of ensemble activity by performing a model-based Bayesian inference sampling procedures. They use this to effectively infer ensemble composition as well as the activity in each ensemble e.g the numbers of ensembles and levels of synchrony. They demonstrate the robustness of this algorithm in a simple simulation model, benchmark it against a prominent existing technique as well as apply it to imaging data in fish and mouse as well neuropix neurophysiological data. The method captures known structure in the data in the data as well as revealing interesting new structure. A Bayesian approach to this problem approach to ensemble identification seems long overdue and this paper makes a welcome addition to the literature. The authors also provide code to implement their algorithm making it of immediate relevance to a large community of researchers thus I would like to see the publication of this paper expedited. I have no substantive criticism but below I outline few point clarification and correction.

In the body and the conclusion the authors say “arbitrary definitions of time frames that characterize synchrony are not required because our method considers sustained periods of activity by extending the time window in which an ensemble is active.”

It is not clear whether this is part of the inference algorithm or the HMM model they use to identify the events. Please clarify.

Relatedly, it would been nice to demonstrate, or at least comment on, the the robustness of the method to the way the data is preprocessed.

Equation 2: Please define z explicitly here.

One general point, which I think goes beyond the scope of what is achieved here, is how useful is it to describe neural activity in terms of ensemble dynamics. This idea is pervasive in the neuroscience community but has its roots in the theory and ideas of coupled oscillators. It is interesting the authors find many cells that are not easily be assigned to an ensemble. Could the authors comment on the possibility that this is because the neural dynamics are not well described by ensemble dynamics.

**Have all data underlying the figures and results presented in the manuscript been provided?**

Reviewer #1: Yes

Reviewer #2: No: The authors indicated that software and data will be made available in a public repository after the manuscript has been accepted for publication.

Reviewer #3: Yes

PLOS authors have the option to publish the peer review history of their article (what does this mean?). If published, this will include your full peer review and any attached files.

Reviewer #1: No

Reviewer #2: No

Reviewer #3: No

---

## [Decision Letter · Decision Letter 1]

9 Oct 2019

Dear Dr Diana,

We are pleased to inform you that your manuscript 'Bayesian inference of neuronal assemblies' has been provisionally accepted for publication in PLOS Computational Biology.

In the meantime, please log into Editorial Manager at https://www.editorialmanager.com/pcompbiol/, click the "Update My Information" link at the top of the page, and update your user information to ensure an efficient production and billing process.

One of the goals of PLOS is to make science accessible to educators and the public. PLOS staff issue occasional press releases and make early versions of PLOS Computational Biology articles available to science writers and journalists. PLOS staff also collaborate with Communication and Public Information Offices and would be happy to work with the relevant people at your institution or funding agency. If your institution or funding agency is interested in promoting your findings, please ask them to coordinate their releases with PLOS (contact ploscompbiol@plos.org).

Thank you again for supporting Open Access publishing. We look forward to publishing your paper in PLOS Computational Biology.

Sincerely,

Claus C Hilgetag

Methods Editor

PLOS Computational Biology

Reviewer's Responses to Questions

**Comments to the Authors:**

Reviewer #1: The authors have adequately addressed all my concerns.

Congratulations.

**Have all data underlying the figures and results presented in the manuscript been provided?**

Reviewer #1: Yes

PLOS authors have the option to publish the peer review history of their article (what does this mean?). If published, this will include your full peer review and any attached files.

Reviewer #1: Yes: Germán Sumbre

---

## [Editor Report · Acceptance letter]

23 Oct 2019

PCOMPBIOL-D-19-00944R1 

Bayesian inference of neuronal assemblies

Dear Dr Diana,

I am pleased to inform you that your manuscript has been formally accepted for publication in PLOS Computational Biology. Your manuscript is now with our production department and you will be notified of the publication date in due course.

With kind regards,

Laura Mallard
